# LLM Distillation for Efficient Few-Shot Multiple Choice Question Answering

## Abstract

Multiple Choice Question Answering (MCQA) is an important problem with numerous real-world applications, such as medicine, law, and education. The high cost of building MCQA datasets makes few-shot learning pivotal in this domain. While Large Language Models (LLMs) can enable few-shot learning, their direct application in real-world scenarios is often hindered by their high computational cost. To address this challenge, we propose a simple yet effective approach that uses LLMs for data generation and scoring. Our approach utilizes LLMs to create MCQA data which contains questions and choices, and to assign probability scores to the generated choices. We then use the generated data and LLM-assigned scores to finetune a smaller and more efficient encoder-only model, DeBERTa-v3-base by leveraging distillation loss. Extensive experiments on the Massive Multi-task Language Understanding (MMLU) benchmark demonstrate that our method improves accuracy from 28.9% to 39.3%, representing a gain of over 10% compared to a baseline finetuned directly on 5-shot examples. This shows the effectiveness of LLM-driven data generation and knowledge distillation for few-shot MCQA.

## 1 Introduction

Multiple Choice Question Answering (MCQA) is a crucial task in natural language understanding with wide applications across domains like medicine (Jin et al., 2021), law (Zheng et al., 2021), and education (Liang et al., 2018). While transformer encoders offer high performance (Vaswani, 2017; Devlin, 2018; Liu, 2019), they typically need a large amount of labeled data to achieve good performance, which can be both expensive and difficult to get (Welbl et al., 2017; Yu et al., 2024). While large language models (LLMs) perform well in few-shot learning (Brown, 2020; Ouyang et al., 2022), their large size makes them difficult to use in real-world scenarios with limited resources. This work addresses the critical need for efficient few-shot MCQA methods by exploring the potential of LLMs to generate training data for more compact encoder-only models.

Despite their few-shot learning capabilities (Brown, 2020; Ouyang et al., 2022), the growing size of LLMs, including powerful open-source variants (Touvron et al., 2023; Team et al., 2024), makes their direct deployment for MCQA costly and impractical in many real-world settings. This motivates the exploration of LLMs not just as tools for answering questions directly, but for generating high-quality, task-specific datasets, as demonstrated in recent studies on classification (Chung et al., 2023) and instruction tuning (Li et al., 2023). While leveraging LLM-generated data for training smaller models shows promise, it often leads to suboptimal performance. This highlights the importance of developing methods that can effectively leverage LLM-generated data to boost the performance of more efficient encoder-only models.

To address the challenges of few-shot MCQA, we propose a novel framework that leverages the strengths of both LLMs and efficient encoder-only models. Our approach begins by generating synthetic MCQA datasets using an open-source LLM, promoting reproducibility and reducing reliance on costly labeled data. We explore two distinct data generation strategies: (1) direct generation in JSON format and (2) a decomposed approach that separates question, positive answer, and negative answer generation, offering flexibility and avoiding format-specific parsing issues. Critically, we further enhance the student encoder-only model by employing LLM-based distillation. Specifically, we use the LLM to give a probability score to the generated answer choices, providing soft labels

that are incorporated into the student model's training through a distillation loss. This combined approach offers a simple, yet effective solution for achieving strong performance in few-shot MCQA scenarios.

Extensive experiments on the Massive Multitask Language Understanding (MMLU) benchmark demonstrate the effectiveness of our proposed framework. Our approach significantly boosts the performance of an encoder-only baseline model, DeBERTa-base-v3 trained with only a 5-shot examples, achieving a 10.4 absolute improvement in accuracy, from 28.9% to 39.3%. Remarkably, applying LLM distillation enables DeBERTa-base-v3 to surpass the few-shot MCQA performance of significantly larger models on the MMLU benchmark. These larger models include the LLaMA-7B base, which achieved 35.1% accuracy, and Flan-T5-250M, which reached 35.9% accuracy after being fine-tuned on a massive multi-task dataset. This highlights the potential of our method to achieve strong MCQA performance with smaller, and more efficient models.

## 2 RELATED WORKS

**Multiple Choice Question Answering (MCQA) Data Generation**. Generating synthetic data for MCQA has been explored previously (Singh Bhatia et al., 2013; Araki et al., 2016), often relying on external resources like Wikipedia (Rodriguez-Torrealba et al., 2022) or knowledge graphs (Yu et al., 2024). While recent work has investigated using LLMs for zero-shot MCQA data generation (Cheung et al., 2023), these approaches typically involve human supervision to ensure quality, limiting the scalability of data creation (Kıyak & Emekli, 2024). In contrast, our work focuses on leveraging LLMs to generate large-scale MCQA datasets automatically, with the aim of distilling their knowledge into efficient encoder-only models for few-shot learning.

**Few-Shot Multiple Choice Question Answering (MCQA)**. Few-Shot MCQA remains a challenging problem, as achieving strong performance often requires large, computationally expensive language models (Anil et al., 2023b; Touvron et al., 2023; Achiam et al., 2023; Anil et al., 2023a). While efficient encoder-only models have shown promise (Sileo, 2024), they typically rely on extensive multi-task training with hundreds of datasets. However, acquiring large-scale MCQA datasets can be costly and time-consuming (Welbl et al., 2017; Yu et al., 2024). In this work, we aim to enable effective few-shot MCQA with encoder-only models by leveraging LLM-generated data and knowledge distillation, addressing the limitations of both data scarcity and computational cost.

**LLM Distillation**. LLM distillation aims to transfer knowledge from large language models into smaller, more efficient ones (Hinton et al., 2015; Xu et al., 2024). A common approach involves generating training data with LLMs and then fine-tuning smaller models on this data. This approach has proven successful in various tasks like classification (Chung et al., 2023), instruction following (Li et al., 2023), and more (Chen et al., 2022; Yehudai et al., 2024; Long et al., 2024). However, most research focuses on distilling into smaller but similar language models (Gu et al., 2024), primarily by creating synthetic datasets (Kim & Rush, 2016; Agarwal et al., 2024). Directly distilling LLM representations is challenging (Xu et al., 2024), and distilling into different model architectures, such as encoder-only models, remains largely unexplored. This gap is particularly pronounced in the context of few-shot MCQA, where the potential of distilling LLMs into encoder-only models remains largely unexplored. While some work has investigated LLM distillation for other tasks, such as semantic search (Liao et al., 2024), to our knowledge our work is the first to systematically explore a combined approach of data generation and probability score-based distillation for enhancing encoder-only models specifically for few-shot MCQA.

## 3 METHOD

Our method addresses few-shot MCQA by leveraging the power of LLMs to generate synthetic training data and then distilling their knowledge into a smaller, more efficient encoder-only model, such as DeBERTa. An overview of our method can be seen in Figure 1. We first generate an MCQA dataset and obtain probability scores for each answer choice using the LLM. These scores serve as soft targets to guide the training of the encoder model, which learns from both the generated data and the distilled LLM knowledge through distillation losses. This approach enables us to enhance the performance of the encoder model in few-shot scenarios while reducing the computational cost associated with deploying compute-intensive LLMs.

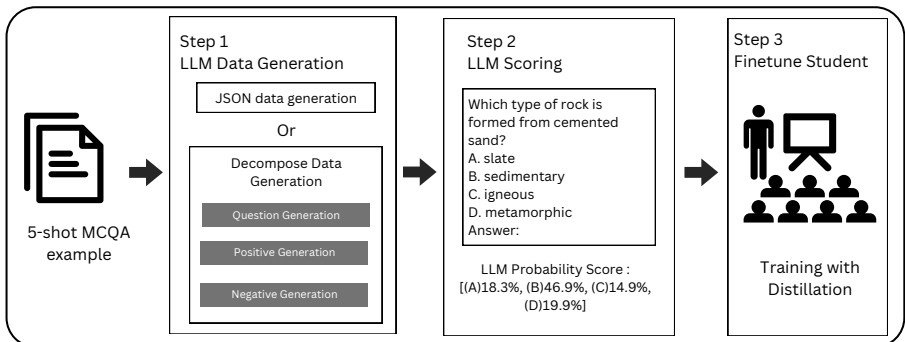

Figure 1: Framework for Few-Shot MCQA using LLM-Generated Data and Distillation.

### 3.1 LLM DATA GENERATION FOR MCQA

Generating high-quality training data is crucial for effective few-shot MCQA. In this subsection, we explore two distinct strategies for leveraging LLMs to create synthetic MCQA datasets: (1) direct generation in JSON format, and (2) a decomposed approach that separates question, positive answer, and negative answer generation. While the direct JSON approach can potentially yield higher-quality data when successful, it can also suffer from parsing issues that reduce the amount of usable data. The decomposed approach, however, avoids the potential parsing issues associated with the JSON method by generating data in a simpler, unstructured format. We detail both methods below and empirically evaluate their impact on the student model's performance in Section 4. We also include all the prompts we used in Appendix E.

#### 3.1.1 JSON

In our first approach, we attempt to directly generate MCQA data in JSON format using few-shot examples. The JSON structure includes the question (string), choices (array of strings), and the answer (integer representing the index of the correct choice). This format implicitly requires the LLM to generate the question first, followed by the answer choices, and finally, the index of the correct answer. However, our experiments reveal that this structured generation process can be challenging for LLMs. They may not consistently adhere to the strict JSON format, leading to parsing errors and a reduction in the amount of usable data. To address this limitation, we propose a decomposed generation method that bypasses the need for parsing JSON output.

#### 3.1.2 DECOMPOSE

Our second approach termed the decomposed generation method, breaks down the MCQA data generation process into three distinct stages: question generation, positive answer generation, and negative answer generation. For each stage, we utilize a few-shot dataset containing questions, positive answers, negative answers, and relevant topics. This decomposition eliminates the need for complex parsing of LLM output, which can be prone to errors when enforcing structured formats like JSON. While this approach might potentially lead to a slight decrease in individual data point quality, it significantly reduces data loss due to parsing failures, ultimately yielding a larger volume of usable training data. For simplicity, we focus on generating data within a single topic, such as high school programming or abstract algebra, ensuring readily available background information. We leverage the few-shot examples and topic information to guide the LLM in generating new MCQA instances.

**Question Generation**. The first stage of the decomposed generation method focuses on creating new questions. We prompt the LLM with instructions like "Create a question about {topic}!", where {topic} is replaced with the chosen subject (e.g., high school programming). To guide the LLM and ensure the generated questions are relevant and similar in style to the target domain, we provide a few-shot prompt consisting of examples randomly sampled from the few-shot dataset. We also adjust the LLM's temperature parameter during this stage to encourage diversity in the generated questions and prevent overfitting to the provided examples.

**positive answer generation**. The second stage focuses on generating the correct answers (positive examples) for the questions created in the previous stage. Similar to question generation, we employ few-shot prompting to guide the LLM. We provide examples of questions and their corresponding correct answers from the few-shot dataset. Then, we present the newly generated questions to the LLM, prompting it to generate relevant and accurate positive answers based on the provided context and examples.

**negative answer generation**. The final stage of data generation focuses on creating plausible but incorrect answer choices (negative examples) for each question. We use few-shot prompting to guide the generation process. To ensure diversity, we generate N negative examples sequentially for each question, prompting the LLM in each iteration to produce a distinct answer, considering all previously generated ones. This iterative approach helps create a diverse set of negative examples for each MCQA instance.

## 3.2 LLM DISTILLATION

After generating the MCQA dataset, we train an encoder model $E_\theta$ on this data. The encoder model comprises a pre-trained encoder, which maps strings to vector representations, followed by a linear layer that outputs scalar values. For each choice $c \in C$ associated with a question, we concatenate the question and the choice and feed it into the encoder, obtaining the output $\hat{y}_c^{enc} \in \mathbb{R}$. We train $E_\theta$ using the standard cross-entropy loss :

$$L_{CE}(p, \hat{p}) = -\frac{1}{C} \sum_{c=1}^{C} p_c log(\hat{p}_c)$$

where $\hat{p}_c$ denotes the model's predicted probability for choice c, and $p_c$ is the corresponding ground truth probability, which is a one-hot vector indicating the correct answer. We then define the loss function $L_{generate}$ for training the encoder model using the generated positive answers as labels. This loss function is given by $L_{generate} = L_{CE}(p, \hat{p})$, where $\hat{p}_c$ is computed using softmax as:

$$\hat{p}_c = \frac{\exp(\hat{y}_c^{enc})}{\sum_{c'=1}^{C} \exp(\hat{y}_{c'}^{enc})}.$$

**Label scoring**. We employ an LLM to score each question and its associated choices, following the approach described in (Robinson & Wingate, 2023). We present the question and all choices to the LLM, with each choice uniquely indexed using characters (e.g., A, B, C). The prompt is designed to elicit a single character as the LLM's output, representing its predicted answer. We record the LLM's score for each unique character, denoted as $\hat{y}_c^{LLM}$, where $c$ represents a choice $c \in C$ associated with a question.

The LLM score $\hat{y}_c^{LLM}$ represents the likelihood of the LLM generating the unique character corresponding to choice c, given the question and all answer choices with their identifiers. Formally:

$$\hat{y}_c^{LLM} \propto P_C(c \mid x),$$

where x is the input string containing the question and all answer choices, each marked with its unique identifier. This scoring method has been shown to improve LLM performance on MCQA tasks (Robinson & Wingate, 2023).

**Training using distillation loss**. We leverage the LLM scores to guide the training of the encoder model through distillation loss. Following the original distillation framework (Hinton et al., 2015), we define the distillation loss as $L_{distill} = L_{CE}(p, \hat{p})$ where p represents the soft target probabilities derived from the LLM scores:

$$p_c = \frac{\exp(\hat{y}_c^{LLM})}{\sum_{c'=1}^{C} \exp(\hat{y}_{c'}^{LLM})}.$$

and $\hat{p}$ represents the encoder model's predicted probabilities, as previously defined. By using the LLM's soft target probabilities as a guide, the distillation loss encourages the encoder model to learn a similar probability distribution over the answer choices, effectively transferring knowledge from the LLM to the smaller encoder model.

| Method | Model Size | STEM | Social Science | Humanities | Other | Average |
|---|---|---|---|---|---|---|
| LLaMA-7B † | 7B | 34.0 | 30.5 | 38.3 | 38.1 | 35.1 |
| Flan-T5-250M † | 248M | 30.1 | 44.0 | 33.9 | 38.9 | 35.9 |
| Gemma-2-2b-it | 2B | 46.8 | 66.9 | 61.6 | 61.3 | 57.7 |
| Llama-3.1-8B-Instruct | 8B | **58.4** | **75.2** | **71.3** | **70.0** | **67.5** |
| DeBERTa 5-shot | 184M | 28.7 | 27.1 | 29.8 | 29.8 | 28.9 |
| Decompose generate | 184M | 27.5 | 36.7 | 35.3 | 35.4 | 33.1 |
| JSON generate | 184M | 28.5 | 36.2 | 35.6 | 35.4 | 33.3 |
| Decompose distill | 184M | 31.6 | 42.4 | 42.6 | 40.3 | 38.4 |
| JSON distill | 184M | **32.5** | **43.2** | **44.3** | **40.6** | **39.3** |
| Tasksource | 184M | 35.6 | 55.4 | **54.4** | 50.8 | 47.5 |
| Tasksource + decompose | 184M | 36.6 | 55.1 | 51.9 | 49.1 | 46.8 |
| Tasksource + JSON | 184M | **37.2** | **56.3** | 54.1 | 50.1 | **48.0** |

Table 1: 5-Shot MCQA Performance on the MMLU Benchmark. Results for LLaMA-7B and Flan-T5-250M (marked with †) are taken from the original papers, which may have different training setups.

## 4 EXPERIMENTS

We conduct experiments to evaluate the effectiveness of our proposed framework for few-shot MCQA. We use a dataset consisting of only 5 MCQA examples covering the same topic, employing Llama-3.1-8B-Instruct [1] as the LLM for data generation and scoring, and DeBERTa-base-v3 (184M parameters) [2] as the efficient encoder-only student model. We chose DeBERTa-base-v3 due to its strong performance and relatively small size, making it suitable for resource-constrained scenarios.

We train the DeBERTa-base-v3 model for 500 iterations with a learning rate of 1e-5, using a batch size of 4 and gradient accumulation for 2 steps, which is equal to using a batch size of 8. For the decompose generation method we set the number of negative examples to be 5 for all experiments, except explicitly mentioned. We average the results across 5 different random seeds for all experiments. Unless otherwise specified, we generate 1024 MCQA examples from the initial 5-shot dataset for training using the temperature of 2. We first evaluate our approach on the MMLU benchmark in Section 4.1 and then conduct an ablation study on the ARC datasets in Section 4.2 to analyze the impact of different components of our method.

### 4.1 MMLU BENCHMARK

We evaluate our approach on the Massive Multitask Language Understanding (MMLU) benchmark (Hendrycks et al., 2020), a widely used benchmark for assessing few-shot MCQA performance in LLMs. MMLU comprises 57 datasets covering diverse topics, each divided into development (dev), validation, and test splits. We utilize only the 5-shot dev set for data generation in all our experiments.

On the MMLU benchmark, we evaluate both the JSON and decomposed data generation methods, both with and without knowledge distillation from the LLaMA-3.1-8B-Instruct model (Dubey et al., 2024). Our evaluation includes comparisons against a range of baselines, including the LLaMA-3.1-8B-Instruct teacher model itself, smaller LLMs like LLaMA-7B (Touvron et al., 2023) and Gemma-2-2B-it (Team et al., 2024) [3], the encoder-decoder model Flan-T5-base (Chung et al., 2024), and a strong encoder-only model, Tasksource DeBERTa-base [4], which was fine-tuned on a large multi-task dataset (Sileo, 2024).

---

[1]https://huggingface.co/meta-llama/Meta-Llama-3.1-8B-Instruct

[2]https://huggingface.co/microsoft/deberta-v3-base

[3]https://huggingface.co/google/gemma-2-2b-it

[4]https://huggingface.co/sileod/deberta-v3-base-tasksource-nli

Table 1 presents the few-shot MCQA results on the MMLU benchmark. As expected, directly training DeBERTa-base-v3 on only 5 examples yields near-random accuracy because of overfitting. Using LLM-generated data significantly improves performance, with the decomposed and JSON methods achieving average gains of 4.2 and 4.3 points, respectively. However, incorporating LLM-generated soft labels via distillation leads to even more substantial improvements, boosting accuracy by an additional 5.3 points for the decomposed method and 6 points for the JSON method. This suggests that while LLMs may generate some incorrect answers during data creation, the distillation process allows them to effectively relabel these instances, leading to a more accurate training signal for the student model. This observation aligns with findings in (Robinson & Wingate, 2023), which demonstrate that framing answer generation as a multiple-choice task can enhance LLM performance.

Our distilled DeBERTa-base-v3 model with 184M parameters achieves encouraging results. It approaches the performance of significantly larger models like LLaMA-7B which is over 30 times larger and Flan-T5-250M, which was extensively fine-tuned on a multi-task dataset. While our method does not yet reach the level of instruction-tuned LLMs like Gemma-2-2B-it and the teacher model LLaMA-3.1-8B-Instruct (as expected, given their substantial size, pretraining data, and instruction tuning), these findings highlight the potential of our approach to achieve strong performance with smaller, more efficient models, making it particularly attractive for resource-constrained settings.

Although our method currently lags behind the Tasksource DeBERTa-base model, which was trained on a massive multi-task dataset including MMLU, our data generation and distillation techniques hold the potential to further boost its performance. Fine-tuning Tasksource DeBERTa-base with our JSON-generated data and distillation results in a 0.5-point average improvement. Interestingly, fine-tuning with data from the decomposed method leads to a performance decrease, indicating that the pre-trained Tasksource model may be more sensitive to data quality and favors the higher-quality data generated by the JSON approach, which benefits from an implicit filtering mechanism.

Considering that our approach generates only 1,024 training instances from a mere 5 initial examples, the observed performance gains suggest that our method effectively distills knowledge from the significantly larger LLaMA-3.1-8B-Instruct model into the smaller DeBERTa-base-v3. While these results are promising, they also highlight opportunities for further research and improvement, such as exploring more advanced data generation and distillation techniques to further bridge the gap with state-of-the-art models.

## 4.2 ABLATION STUDY

In this section, we conduct an ablation study on the ARC-easy and ARC-Challenge benchmarks (Clark et al., 2018) to analyze the impact of different components of our proposed method. We use a 5-shot learning setup, randomly selecting 5 examples from the training set to generate 1024 data points, which are then scored using LLaMA 3.1-8B-Instruct. We train DeBERTa-base-v3 models on the generated data and scores.

We investigate several key aspects. First, we examine the effect of the number of generated data points, as detailed in Section 4.2.1. Second, in Section 4.2.2, we analyze the impact of using smaller LLMs for data generation and scoring. This section also includes a comparison with a paraphrasing baseline. Finally, we explore the influence of the temperature hyperparameter during data generation in Section 4.2.3.

### 4.2.1 EFFECT OF NUMBER OF GENERATED DATA

To analyze the impact of the number of generated data points, we evaluate models trained on datasets of varying sizes: [16, 32, 64, 128, 256, 512, 1024]. We use a cumulative approach, where each larger dataset includes all the data points from the smaller datasets. For instance, the 32-sample dataset consists of the initial 16 samples plus 16 new samples. This ensures that any observed performance changes can be directly attributed to the increase in training data. We compare the performance of models trained on: (1) real data from the ARC training set, (2) generated data, and (3) generated data augmented with LLM distillation.

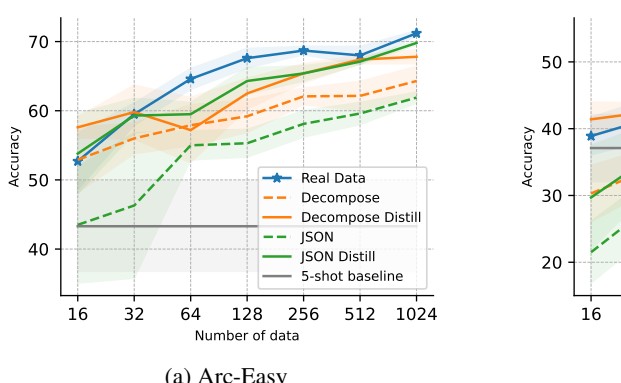
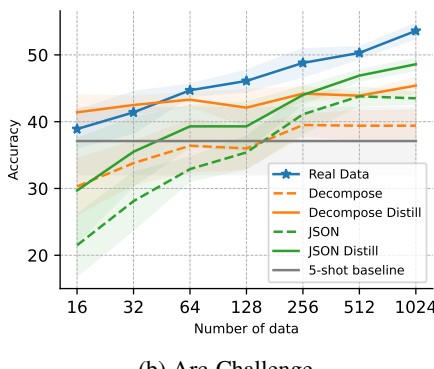

(a) Arc-Easy

(b) Arc-Challenge

Figure 2: Effect of Generated Data Size on Few-Shot MCQA Accuracy. The figure compares the performance of DeBERTa-base-v3 trained on varying amounts of generated data (using both JSON and Decompose methods), with and without LLM distillation, against a baseline trained on real data from the ARC-Easy (a) and ARC-Challenge (b) datasets.

| Generation Method | ARC-Easy | | | ARC-Challenge | | |
|---|---|---|---|---|---|---|
| | Generate | Distill | SR | Generate | Distill | SR |
| LLaMA-3.1-8B JSON | $61.9 \pm 0.8$ | $\mathbf{69.8 \pm 0.3}$ | 0.52 | $\mathbf{43.6 \pm 0.9}$ | $\mathbf{48.6 \pm 0.9}$ | 0.66 |
| Gemma-2-2b JSON | $50.6 \pm 4.9$ | $68.2 \pm 0.6$ | 0.21 | $40.1 \pm 2.6$ | $48.0 \pm 0.7$ | 0.28 |
| LLaMA-3.1-8B Decomp. | $\mathbf{64.3 \pm 2.1}$ | $67.8 \pm 1.0$ | $\mathbf{1.0}$ | $39.4 \pm 2.2$ | $45.3 \pm 1.1$ | $\mathbf{1.0}$ |
| Gemma-2-2b Decomp. | $61.6 \pm 2.4$ | $60.0 \pm 1.5$ | $\mathbf{1.0}$ | $37.7 \pm 2.2$ | $43.9 \pm 1.2$ | $\mathbf{1.0}$ |
| Paraphrase | $52.8 \pm 3.0$ | $42.2 \pm 8.6$ | $\mathbf{1.0}$ | $36.6 \pm 3.1$ | $41.8 \pm 2.5$ | $\mathbf{1.0}$ |

Table 2: Impact of Generation and Scoring Methods on Performance. The table shows the accuracy of different language models on ARC-Easy and ARC-Challenge datasets, using various generation and scoring methods. "SR" denotes the success rate of JSON parsing. "Decomp." indicates a decomposition-based generation method. All models utilize instruction-tuned versions.

Figure 2 presents the results of this analysis. We observe that both data generation and LLM distillation are crucial for improving performance in the few-shot setting. Training DeBERTa-base-v3 with only 5 examples leads to high variance across different random seeds, indicating instability due to limited data. Leveraging generated data substantially improves performance and reduces variance. Additionally, the inclusion of LLM distillation further boosts accuracy and reduces variance, demonstrating the complementary benefits of these techniques. Our approach significantly outperforms the 5-shot baseline, demonstrating its effectiveness in leveraging limited data for few-shot MCQA.

We generally observe increasing performance with larger amounts of generated data, particularly when combined with LLM distillation. Notably, LLM distillation consistently boosts accuracy across all data sizes and generation methods, demonstrating its robustness and effectiveness. Although our method does not surpass the performance of a model trained on abundant real data, achieving comparable results with significantly less real data is significant. Using JSON-generated data and distillation, we achieve accuracy similar to training on 512 real samples for ARC-Easy and 256 real samples for ARC-Challenge. This highlights the potential of our approach to reduce the reliance on extensive, expensive real-world MCQA datasets.

### 4.2.2 Effect of Generation and Scoring Method

We further investigate the influence of the LLM used for data generation and scoring. Table 2 presents the results for data generated by LLaMA-3.1-8B-Instruct and the smaller Gemma-2-2B-it. Interestingly, our approach achieves comparable performance with both LLMs, suggesting that even smaller LLMs can effectively generate and score data for our framework. Notably, using the JSON

| Generation Method | Temperature | ARC-Easy | | | ARC-Challenge | | |
|---|---|---|---|---|---|---|---|
| | | Generate | Distill | SR | Generate | Distill | SR |
| Decompose | 0.5 | $60.9 \pm 1.6$ | $59.7 \pm 2.1$ | **1.0** | $34.1 \pm 4.9$ | $37.9 \pm 4.1$ | **1.0** |
| | 1.0 | $63.2 \pm 1.6$ | $66.4 \pm 1.1$ | **1.0** | **$39.6 \pm 2.2$** | $41.9 \pm 1.7$ | **1.0** |
| | 2.0 | **$64.3 \pm 2.1$** | **$67.4 \pm 0.4$** | **1.0** | $39.4 \pm 2.2$ | **$45.4 \pm 1.2$** | **1.0** |
| JSON | 0.5 | $50.5 \pm 4.1$ | $54.9 \pm 8.0$ | **1.0** | $36.7 \pm 1.9$ | $34.6 \pm 4.4$ | **1.0** |
| | 1.0 | $61.5 \pm 1.2$ | $65.1 \pm 0.9$ | 0.99 | $41.9 \pm 3.1$ | $41.7 \pm 2.3$ | **1.0** |
| | 2.0 | **$61.9 \pm 0.8$** | **$69.8 \pm 0.3$** | 0.52 | **$43.6 \pm 0.9$** | **$48.6 \pm 0.9$** | 0.66 |

Table 3: Effect of Generation Temperature on Few-Shot MCQA Performance. The table compares the performance of the Decompose and JSON generation methods, with and without distillation, across different temperature settings. SR denotes the success rate of JSON parsing.

generation method with both LLMs yields similar results, although the success rate of JSON parsing varies significantly. We hypothesize that this is because the JSON format acts as an implicit filter, discarding poorly formatted data, which is more likely to occur with the smaller Gemma model. Furthermore, we observe that distillation consistently improves performance across all LLM and generation method combinations, indicating its ability to refine potentially noisy labels from the generated data.

We also compare our method to a baseline that uses paraphrasing to augment the 5-shot data. We show the prompt we use in Appendix E. While paraphrasing has proven effective for various NLP tasks (Feng et al., 2021), our results demonstrate that LLM-based data generation is significantly more effective for few-shot MCQA. We use LLaMA-3.1-8B-Instruct to paraphrase the questions and choices in the 5-shot dataset via few-shot prompting. Even when using data generated by the smaller Gemma-2-2B-it model, our approach substantially outperforms the paraphrasing baseline on both ARC datasets. This highlights the importance of generating new data, rather than simply rephrasing existing examples, to enhance data diversity and improve performance in few-shot settings.

### 4.2.3 EFFECT OF GENERATION HYPERPARAMETERS

We now analyze the influence of the temperature hyperparameter, which controls the diversity of the generated data, on the performance of our approach. Table 3 presents the results for both the JSON and decomposed generation methods across different temperature settings. We observe that temperature plays a crucial role, and increasing it generally leads to improved performance. This highlights the importance of data diversity for effective few-shot MCQA, demonstrating that even simple techniques like temperature control can significantly impact the quality of the generated data.

While increasing the temperature doesn't always consistently improve performance when using only the generated data, the benefits become much more pronounced when combined with distillation loss. We hypothesize that this is because LLMs can introduce noise into the generated data, and distillation helps mitigate this noise by encouraging the student model to learn a smoother probability distribution over the answer choices, similar to label smoothing, which has been shown to improve robustness to noisy labels (Szegedy et al., 2016; Lukasik et al., 2020). To further investigate this, we experimented with replacing the soft labels from the LLM with hard labels (choosing the most probable answer) but observed inferior performance compared to using the full probability distribution, we provide the results in Appendix C.2. This highlights the importance of leveraging the soft labels provided by the LLM for effective knowledge distillation.

While the JSON generation method can yield better performance at higher temperatures, it often comes at the cost of a lower usable data rate due to parsing errors. Many generated instances must be discarded because they don't adhere to the strict JSON format. In contrast, the decomposed method consistently achieves competitive performance without requiring any parsing. Even when reducing the temperature for JSON generation to 1 to improve the parsing success rate, its performance still falls short of the decomposed method. This demonstrates that the decomposed approach offers a more robust and efficient alternative.

For the decomposed generation method, we also investigated the effect of varying the number of negative examples generated per question. The results, presented in Appendix Table 6, demonstrate that our method is robust to changes in this parameter. We did not perform this ablation study for the JSON generation method because it does not allow for controlling the number of choices.

## 5 LIMITATIONS

Our work relies on a robust, instruction-tuned LLM, which is currently readily available in English but might be less accessible in other languages. This language dependence, coupled with the reliance on strong LLM capabilities, could limit the generalizability of our method to scenarios where suitable LLMs are unavailable or less powerful.

Despite significant improvements over the naive 5-shot baseline, our method still exhibits a substantial performance gap compared to models trained with extensive data and multi-task learning, as well as the teacher LLM itself. Bridging this gap by exploring more advanced data generation techniques, incorporating diverse knowledge sources, or developing more effective distillation strategies remains a promising direction for future research.

Another limitation of our approach is the potential for bias in the LLM-generated data. LLMs are trained on massive text corpora, which inevitably contain societal biases. These biases can be reflected in the generated questions and choices, potentially leading to a biased downstream encoder model. This inherited bias could result in unfair or discriminatory outcomes when the model is deployed in real-world applications. Mitigating this bias is a crucial area for future work.

A further limitation is that our current work focuses on MCQA tasks with relatively short question-and-answer contexts, which are easier for current LLMs to generate and score effectively. We observed increased noise in the generated data when dealing with longer contexts, evidenced by a performance degradation when fine-tuning the Tasksource DeBERTa-base model on generated data for longer-context MMLU tasks. This suggests that generalizing our approach to tasks involving longer contexts will require further research.

## 6 CONCLUSION

This work demonstrates the effectiveness of leveraging LLMs for both data generation and probability-based distillation to enable strong few-shot MCQA performance in smaller, more efficient encoder-only models. Our approach achieves encouraging results on the MMLU benchmark, even approaching the performance of significantly larger models like LLaMA-7B and Flan-T5-250M, which benefited from more extensive training data and different training objectives. This highlights the potential of our method to achieve strong performance with more compact and computationally efficient models. However, a performance gap remains compared to models trained with large-scale multi-task data, suggesting opportunities for further improvement. Future work will focus on bridging this gap by exploring more advanced data filtering techniques to enhance the quality of the generated data and investigating novel distillation strategies to maximize knowledge transfer from LLMs to smaller models. Additionally, extending our approach to effectively handle longer-context MCQA tasks is a crucial direction for future research.

### REPRODUCIBILITY STATEMENT

We are committed to ensuring the reproducibility of our work. The code for our data generation and model training pipeline will be made publicly available upon publication. We have provided detailed descriptions of our methodology, including the specific LLMs and encoder models used, the data generation strategies, the distillation process, and the evaluation setup. Furthermore, we have included the full results of our experiments, both aggregated and detailed, in the main paper and the appendix. We believe that these efforts will enable other researchers to reproduce our findings and build upon our work to further advance the field of few-shot MCQA.

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

## A  FUTURE WORKS

This work lays the foundation for several promising research directions, with the potential to significantly advance efficient few-shot learning in multiple choice question answering and beyond. Specifically, we identify the following key areas for future exploration:

**Advanced Distillation Techniques.** In this work, we used a simple distillation approach to establish a clear baseline. Exploring more sophisticated distillation techniques, such as sequence-level knowledge distillation, attention-based distillation, or other distillation approach, that could further enhance performance.

**Benchmark Dataset Creation.** Our findings suggest that the JSON generation method coupled with LLM distillation is a promising approach for creating high-quality MCQA data. This method appears to act as an effective filter for selecting higher-quality generated examples. Combining our approach with automated quality filtering based on perplexity or LLM-based scoring, post-processing techniques to refine generated text, and retrieval-augmented generation to incorporate external knowledge could facilitate the creation of valuable benchmark datasets for few-shot MCQA. This would require developing robust filtering and evaluation metrics to ensure the quality and diversity of the generated datasets.

**Improving Decomposed Generation.** While the decomposed generation method offers advantages in terms of data generation efficiency, it can produce noisy data due to longer and less structured answers. Investigating more sophisticated prompting techniques could mitigate this limitation. Incorporating constraints into the prompts, specifying the desired length or format of the answers, could improve the quality of the generated data. Iterative refinement, where feedback is provided to the LLM to revise its responses, is another promising avenue. Additionally, using more diverse and representative examples in the few-shot prompts could guide the LLM towards generating more appropriate answers.

**Applications Beyond MCQA.** Our framework has broader applicability beyond MCQA. Within NLP, it could be applied to tasks like text classification, sequence tagging, or any task where efficient few-shot learning is desirable. In these applications, the LLM could generate synthetic training examples and provide soft labels or confidence scores to guide the training of a smaller model. Furthermore, with the advancements in Vision-Language Models (VLLMs), our approach could be extended to vision tasks. For example, in Visual Question Answering (VQA), the VLLM could generate image captions, which could then be used to synthesize images with a generative model. The VLLM could also generate the question and possible answers. The generated VQA data, along with the VLLM's confidence scores for each answer, can then be used to distill the knowledge into a smaller, more efficient vision-language model or even a specialized VQA architecture.

## B  IMPLEMENTATION DETAILS

We implemented our method using the Transformers library (Wolf et al., 2020) for loading and interacting with the LLMs and encoder models, and the Datasets library (Lhoest et al., 2021) for loading and processing the datasets. This appendix provides detailed information about the computational resources, data generation process, and model training procedure

### B.1  COMPUTATION RESOURCES

The experiments were conducted using three machines:

- Two machines: AMD Ryzen 5 2600 Six-Core Processor, NVIDIA RTX 3090 24GB GPU.
- One machine: AMD Ryzen Threadripper 1920X 12-Core Processor, two NVIDIA RTX 3090 24GB GPUs.

### B.2  DATA GENERATION DETAILS

We utilize instruction-tuned LLMs that follow the standard System, User, and Assistant role format. The System role sets the overall instructions for the model's behavior, the User role provides specific

commands or prompts, and the Assistant role generates the model's responses. Our 5-shot prompting approach includes the few-shot examples as the first five User-Assistant interactions. Subsequent User prompts are then used to elicit new responses from the LLM for data generation or scoring.

**JSON Generation:** For the JSON generation method, we use a straightforward 5-shot prompting approach. The full prompt examples for ARC-Easy and ARC-Challenge are shown in Tables 17 and 18, respectively. All five examples in the prompt use the same question, but we shuffle their order to encourage diversity in the generated outputs.

**Decomposed Generation:** The decomposed generation method follows a similar 5-shot prompting structure as the JSON approach. However, we divide the generation process into three distinct stages: (1) question generation, (2) positive answer generation, and (3) negative answer generation. Each stage utilizes a separate prompt, as shown in Tables 19, 20, and 21, respectively.

Most data generation tasks could be run on a single NVIDIA RTX 3090 GPU. However, certain MMLU tasks with longer sequences, such as high_school_european_history, high_school_us_history, high_school_world_history, professional_law, professional_medicine, and security_studies, required two RTX 3090 GPUs to avoid out-of-memory errors.

When using the JSON generation method, we encountered challenges with certain datasets that required significantly longer generation times to obtain 1024 usable data points. This was primarily due to a combination of long sequences and low parsing success rates. The affected datasets and the number of usable data points we were able to obtain are as follows:

- college_mathematics: 512
- formal_logic: 538
- high_school_european_history: 327
- high_school_us_history: 305
- high_school_world_history: 765

After generating the MCQA data (questions, choices, and answers), we use the LLM to score each choice. Table 22 shows an example of the scoring prompt, which includes the 5-shot examples and the newly generated data. To obtain the scores, we extract the logits (pre-softmax outputs) corresponding to the unique character identifiers for each choice. To avoid out-of-memory errors during scoring, we limit the prompt length to 1024 tokens when using a single GPU and 3200 tokens when using two GPUs. This is necessary because some generated instances can contain very long sequences

### B.3 MODEL TRAINING DETAILS

We train the DeBERTa-base-v3 model, which takes a question and a choice as input, for all our experiments. The model uses the pooled output of the encoder, which is then fed into a linear layer to produce a scalar output. We train the model using the Adam optimizer (Kingma, 2014) for 500 iterations, with a batch size of 4 and gradient accumulation for 2 steps (effectively a batch size of 8). This allows the model to be exposed to approximately 4000 MCQA examples during training. We use a learning rate of 1e-5.

Before training, we filter the dataset to avoid out-of-memory errors during training. We use the DeBERTa tokenizer to count the number of tokens for the concatenation of each question and its corresponding choices. If the total number of tokens for any question-choice pair exceeds a predefined maximum (max_tokens), we discard that data point. For most experiments, we set max_tokens to 320. However, for MMLU tasks with longer sequences, we increased max_tokens to 480.

## C ADDITIONAL RESULTS

This appendix provides supplementary results and analyses to complement the findings presented in the main paper. It is organized as follows:

| Method | ARC-Easy | ARC-Challenge |
|---|---|---|
| Tasksource | $72.8 \pm 0.0$ | $51.2 \pm 0.0$ |
| Tasksource + JSON distill | $74.5 \pm 0.5$ | $54.7 \pm 1.0$ |
| Gemma-2-2b-it | $89.6 \pm 0.0$ | $73.7 \pm 0.0$ |
| Llama-3.1B-Instruct | $\mathbf{93.3 \pm 0.0}$ | $\mathbf{82.6 \pm 0.0}$ |
| deberta 5-shot baseline | $26.5 \pm 13.8$ | $37.1 \pm 5.0$ |
| Decompose generate | $64.3 \pm 2.1$ | $39.3 \pm 2.2$ |
| Decompose distill | $67.8 \pm 1.0$ | $45.3 \pm 1.1$ |
| JSON generate | $61.9 \pm 0.8$ | $43.6 \pm 0.9$ |
| JSON distill | $\mathbf{69.8 \pm 0.3}$ | $\mathbf{48.6 \pm 0.9}$ |

Table 4: Result on arc-easy and arc-challenge.

- Section C.1: Baseline and Teacher Model Performance on ARC Datasets. This section presents the performance of the 5-shot baseline, the teacher LLMs, and our proposed method on the ARC-Easy and ARC-Challenge datasets.

- Section C.2: Effect of Distillation Temperature. This section examines the impact of varying the temperature of the softmax function during distillation on the performance of the student model.

- Section C.3: Effect of Number of Negative Examples (Decomposed Method). This section analyzes the influence of the number of negative examples generated per question on the performance of the decomposed generation method.

- Section C.4: Effect of Number of Generated Data Points (Numerical Results). This section provides the detailed numerical results corresponding to Figure 2 in the main paper.

- Section C.5: Generation Duration Comparison. This section compares the time required to generate datasets using the decomposed and JSON generation methods.

- Section C.6: Lightweight LLM Comparison. This section compares the memory usage and performance of our method with lightweight LLMs.

- Section C.7: Binary Classification Extensions. This section explores the application of our method to binary classification tasks, such as scoring the correctness of question-answer pairs.

- Section C.8: Learning MCQA Format vs. Domain Knowledge. This section investigates whether our approach primarily teaches the model the MCQA format or if it also improves domain-specific knowledge.

- Section C.9: Dataset Semantic Similarity. This section analyzes the semantic similarity between generated, training, and test set questions to assess data novelty and quality.

- Section C.10: Detailed MMLU Results. This section presents the detailed results for our method on the MMLU benchmark, broken down by subject area.

## C.1 BASELINE AND TEACHER PERFORMANCE ON ARC-E AND ARC-C

Table 4 presents the performance of the 5-shot DeBERTa baseline, the teacher LLMs (LLaMA-3.1B-Instruct and Gemma-2-2b-it), and our proposed method on the ARC-Easy and ARC-Challenge datasets. Consistent with the MMLU results, LLM distillation significantly improves performance over the 5-shot baseline. However, a notable gap remains between our student models and the teacher LLMs, as well as the Tasksource DeBERTa-base model, which benefited from extensive multi-task training. This highlights the potential for further improvement in our approach, particularly in terms of bridging the gap with models trained on larger, more diverse datasets.

| Temperature | ARC-Easy | | ARC-Challenge | |
| --- | --- | --- | --- | --- |
| | Decompose | JSON | Decompose | JSON |
| 0.0 | $64.9 \pm 1.4$ | $66.8 \pm 0.7$ | $42.5 \pm 1.4$ | $46.7 \pm 0.9$ |
| 0.5 | $67.4 \pm 0.5$ | $69.2 \pm 0.9$ | $45.1 \pm 0.7$ | $48.2 \pm 1.0$ |
| 1.0 | $67.8 \pm 1.0$ | $\mathbf{69.8 \pm 0.3}$ | $\mathbf{45.4 \pm 1.2}$ | $\mathbf{48.6 \pm 0.9}$ |
| 1.5 | $67.8 \pm 1.7$ | $69.7 \pm 1.0$ | $44.2 \pm 1.0$ | $45.4 \pm 1.2$ |
| 2.0 | $\mathbf{67.9 \pm 0.5}$ | $69.5 \pm 0.5$ | $45.4 \pm 1.2$ | $48.1 \pm 1.2$ |

Table 5: Effect of the distillation temperatures on generated data.

| Negative Number | ARC-Easy | | ARC-Challenge | |
| --- | --- | --- | --- | --- |
| | Generate | Distill | Generate | Distill |
| 3 | $61.3 \pm 1.6$ | $67.4 \pm 0.4$ | $38.4 \pm 2.2$ | $46.3 \pm 0.1$ |
| 4 | $62.8 \pm 1.5$ | $\mathbf{67.7 \pm 1.3}$ | $38.7 \pm 1.5$ | $47.0 \pm 0.5$ |
| 5 | $62.2 \pm 1.6$ | $67.5 \pm 1.8$ | $37.4 \pm 1.6$ | $46.5 \pm 0.3$ |
| 6 | $\mathbf{62.9 \pm 1.8}$ | $67.2 \pm 0.1$ | $\mathbf{39.3 \pm 1.8}$ | $\mathbf{47.1 \pm 0.5}$ |

Table 6: Effect of the number of negatives in decompose method.

## C.2 STUDENT MODEL DISTILLATION TEMPERATURE

In the distillation process, we can control the temperature of the softmax function applied to both the student model's predictions and the teacher LLM's likelihood scores:

$$p_c = \frac{\exp(\hat{y}_c/r)}{\sum_{c'=1}^{C} \exp(\hat{y}_{c'}/r)}.$$

Where r denotes the temperature. A temperature of 0 is equivalent to using hard labels from the teacher model (selecting the most probable answer). Table 5 presents the results of varying the distillation temperature. We observe that using a temperature of 0 leads to a significant performance drop, highlighting the importance of soft-label distillation for mitigating the impact of noise in the generated data. For distillation temperatures other than 0, we observe that there's no significant difference between temperatures. This shows that our method is robust to the temperature used during distillation.

## C.3 EFFECT OF NUMBER OF NEGATIVE IN DECOMPOSE METHOD

We investigated the effect of varying the number of negative examples generated per question for the decomposed generation method. The results, presented in Table 6, show no significant performance difference across the range of negative examples tested on both ARC-Easy and ARC-Challenge. This suggests that the decomposed method is robust to the number of negative choices used during data generation.

## C.4 EFFECT OF NUMBER OF GENERATED DATA

Table 7 provides the numerical results corresponding to Figure 2, showing the effect of the number of generated data points on model performance. As expected, increasing the amount of generated data generally leads to improved performance, particularly when combined with LLM distillation.

## C.5 GENERATION DURATION COMPARISON

This section analyzes the time required to generate data using LLaMA 3.1-8B-Instruct for a subset of MMLU and ARC datasets. We selected five MMLU datasets: High School European History, High School US History, High School World History, Sociology, and US Foreign Policy. The first three represent tasks with particularly long contexts, which we found to be the most challenging for data

| Dataset | Method | 8 | 16 | 32 | 64 | 128 | 256 | 512 | 1024 |
|---------|--------|-----|-----|-----|-----|-----|-----|-----|------|
| ARC-E | Real data | 31.9 | 52.7 | 59.5 | 64.6 | 67.6 | 68.7 | 68.0 | 71.2 |
| | Decompose generate | 46.3 | 52.9 | 56.0 | 57.9 | 59.2 | 62.1 | 62.1 | 64.3 |
| | Decompose distill | 44.8 | 57.6 | 59.8 | 57.2 | 62.5 | 65.4 | 67.4 | 67.8 |
| | JSON generate | 22.1 | 43.5 | 46.3 | 55.0 | 55.3 | 58.1 | 59.6 | 61.9 |
| | JSON distill | 54.7 | 53.8 | 59.3 | 59.5 | 64.3 | 65.4 | 67.1 | 69.8 |
| ARC-C | Real data | 38.7 | 38.9 | 41.4 | 44.7 | 46.1 | 48.8 | 50.3 | 53.6 |
| | Decompose generate | 32.1 | 30.3 | 33.8 | 36.4 | 36.0 | 39.5 | 39.4 | 39.4 |
| | Decompose distill | 36.8 | 41.4 | 42.5 | 43.3 | 42.1 | 44.2 | 43.9 | 45.4 |
| | JSON generate | 20.8 | 21.5 | 28.1 | 32.9 | 35.4 | 41.1 | 43.8 | 43.6 |
| | JSON distill | 19.9 | 29.7 | 35.5 | 39.3 | 39.3 | 44.0 | 46.9 | 48.6 |

Table 7: Effect of number of generated data againts performance and its comparison with real data

| Dataset Name | Generation Method | Distill Avg | Batch Size | Data Counts | Generate Time(S) | Estimate Total Time(H) |
|--------------|-------------------|-------------|------------|-------------|------------------|------------------------|
| High School European History | Decompose | 49.2 | 4 | **1024** | **31.9** | 9.06 |
| | JSON | **50.8** | **6** | 327 | 90.4 | **8.21** |
| High School US History | Decompose | **50.4** | 4 | **1024** | **19.8** | **5.63** |
| | JSON | 45.5 | **6** | 305 | 74.6 | 6.32 |
| High School World History | Decompose | **53.1** | 4 | **1024** | **11.9** | **3.39** |
| | JSON | 51.5 | **8** | 765 | 32.9 | 6.99 |
| Sociology | Decompose | 48.0 | **10** | **1024** | **3.2** | **0.91** |
| | JSON | **50.3** | 10 | **1024** | 5.0 | 1.43 |
| US Foreign Policy | Decompose | 52.0 | 8 | **1024** | **4.0** | **1.13** |
| | JSON | **53.8** | **10** | **1024** | 5.4 | 1.55 |
| ARC-Easy | Decompose | 67.8 | **4** | **1024** | **3.7** | **1.04** |
| | JSON | **69.8** | **4** | **1024** | 4.3 | 1.24 |
| ARC-Challenge | Decompose | 45.4 | **4** | **1024** | **3.3** | **0.94** |
| | JSON | **48.6** | **4** | **1024** | 3.7 | 1.04 |

Table 8: The comparison of performance and generation time on some subset of MMLU, with also ARC-Easy and ARC-Challenge.

| Sequence Length | DeBERTa-base | LLaMA 1B | LLaMA 1B 4 bit | Gemma 2B | Gemma 2B 4 bit |
|---|---|---|---|---|---|
| 128 | 1.701 | 3.576 | 2.211 | 6.351 | 3.444 |
| 256 | 1.728 | 3.773 | 2.421 | 6.705 | 3.912 |
| 512 | 1.768 | 4.134 | 2.794 | 7.393 | 4.585 |
| 1024 | 2.060 | 4.872 | 3.507 | 8.792 | 5.971 |
| 2048 | 3.152 | 6.235 | 4.870 | 11.610 | 8.699 |
| 4096 | 6.600 | 9.157 | 7.741 | 17.050 | 14.207 |

Table 9: Memory usage comparison of LLMs and encoder only method based on sequence length in GB.

generation and required two GPUs. Sociology, US Foreign Policy, ARC-Easy, and ARC-Challenge are included to provide estimated generation times for more typical tasks.

To maximize GPU utilization, we adjusted the batch size for data generation, noting that larger batch sizes generally lead to faster generation but are constrained by GPU memory. We estimate the time based on generating at least 200 data points. Table 8 shows the chosen batch size for each dataset, along with the number of generated data points, the resulting model performance, the time taken to generate a single data point (in seconds), and the estimated total generation time (in hours).

As shown in Section D, the decomposed method generally requires smaller batch sizes due to the longer sequences it produces. However, despite using smaller batch size and the longer sequences, the decomposed method often achieves faster overall data generation. This is attributed to the lower parsing success rate of the JSON method, which requires generating and then discarding a significant portion of the data. As a result, generating a complete dataset with the JSON method often takes longer. Notably, for datasets with very long sequences and low parsing success rates, the decomposed method can even yield higher performance while using a similar computational budget for data generation. This highlights a key advantage of the decomposed approach: its ability to efficiently generate usable data, even if the individual data points might be of slightly lower quality.

### C.6 Lightweight LLM Comparison

To compare our method with a lightweight LLM, we evaluated the LLaMa-3.2-1B-Instruct and Gemma-2b-it models. We analyzed the memory usage of both LLMs, both with and without 4-bit quantization, and compared them to the encoder-only DeBERTa-base model during inference. We measured memory consumption using the vmlDeviceGetMemoryInfo function from pynvml, feeding each model sequences of 128 to 4096 random tokens. Results are shown in Table 9. We observe that even with a lightweight 1B parameter LLM and 4-bit quantization, the LLM memory usage is still greater than DeBERTa-base. We believe this is caused by the larger activation sizes of LLMs, which are not quantized during inference, thus requiring more memory. LLMs also often require longer input sequences than encoder-only models due to instructions, concatenated choices, and few-shot examples. This can lead to input sequences that are significantly longer, further increasing memory requirements.

Table 10 shows the performance comparison. Quantization reduces LLM performance, as expected. To compare at similar memory footprints, we also evaluated LLMs without few-shot prompting. Performance degrades significantly, particularly for the smaller LLaMa-3.2-1B-Instruct. We observer that the memory usage of DeBERTa is most similar to that of LLaMA-3.2-1B-Instruct with 4-bit quantization. Compared to this model, DeBERTa achieves comparable performance on MMLU, within 1 percentage point. When considering similar sequence lengths during inference, DeBERTa significantly outperforms the 4-bit quantized LLaMA-3.2-1B-Instruct model. This reduced memory footprint and potentially faster inference speed makes DeBERTa a more attractive option for deployment on resource-constrained devices.

| Method | STEM | Social Science | Humanities | Other | Average |
|---|---|---|---|---|---|
| Gemma-2-2b-it(5-shot) | **46.8** | **66.9** | **61.6** | **61.3** | **57.7** |
| Gemma-2-2b-it 4 bit(5-shot) | 45.2 | 64.5 | 59.1 | 57.5 | 55.2 |
| Gemma-2-2b-it 4 bit(0-shot) | 42.6 | 58.6 | 56.2 | 54.9 | 51.9 |
| LLaMA-3.2-1B-Instruct(5-shot) | 36.5 | 47.8 | 46.3 | 45.1 | 43.1 |
| LLaMA-3.2-1B-Instruct 4 bit(5-shot) | 35.7 | 45.6 | 42.2 | 40.6 | 40.3 |
| LLaMA-3.2-1B-Instruct 4 bit(0-shot) | 29.4 | 33.7 | 26.7 | 29.1 | 29.6 |
| DeBERTa-v3 + JSON distill (5-shot) | 32.5 | 43.2 | 44.3 | 40.6 | 39.3 |
| Tasksource + JSON distill(5-shot) | **37.2** | **56.3** | **54.1** | **50.1** | **48.0** |

Table 10: Performance Comparison with small and 4-bit LLMs

| Method | ARC-Easy | ARC-Challenge |
|---|---|---|
| 1024 real data binary | **56.81 ± 1.47** | 40.25 ± 4.08 |
| 5 real data binary | 27.01 ± 10.09 | 14.23 ± 9.64 |
| 1024 JSON binary | 48.86 ± 1.42 | 32.20 ± 6.93 |
| 1024 JSON MCQA heuristic | 49.50 ± 1.35 | **42.38 ± 0.54** |

Table 11: Performance Comparison with small and 4-bit LLMs

## C.7 BINARY CLASS EXTENSIONS

In real-world applications like fact verification or information retrieval, it's often necessary to determine the correctness of a given answer without explicitly presenting choices. This necessitates framing the problem as binary classification. To investigate our framework's applicability to this setting, we consider two approaches. First, we train a model with binary cross-entropy (BCE) loss and sigmoid activation on the final layer, using data generated by the LLaMA-3.1-8B-Instruct using JSON format. Second, we use a simple heuristic approach. We train the model exactly as in the MCQA setting. During the evaluation, we search for a constant threshold using the same data by averaging the logits produced by the model for each question-answer pair across all choices. At inference, if the logit for a pair is above the threshold, the pair is classified as correct.

For evaluation, we use the binary F1-score, as the number of correct and incorrect pairs is not balanced. Results are presented in Table 11. As expected, using only 5-shot examples performs poorly, while training on real binary data achieves good results. Interestingly, models trained in the MCQA setting and then classified using the heuristic approach outperform the models trained directly with BCE loss on generated data. Furthermore, models trained with distillation and then classified using the heuristic demonstrate smaller variance and even outperform models trained on real binary data on the ARC-Challenge dataset. We hypothesize that the heuristic, by leveraging the full probability distribution learned during MCQA training, allows the model to develop a more nuanced representation of correctness.

| Method | STEM | Social Science | Humanities | Other | Average |
|---|---|---|---|---|---|
| Trained on MMLU generated | **32.5** | **43.2** | **44.3** | **40.6** | **39.3** |
| Trained on Arc-E 5-shot | 22.0 | 22.8 | 21.9 | 22.5 | 22.3 |
| Trained on Arc-E generated | 32.3 | 40.5 | 41.4 | 40.3 | 37.9 |

Table 12: Performance Comparison with small and 4-bit LLMs

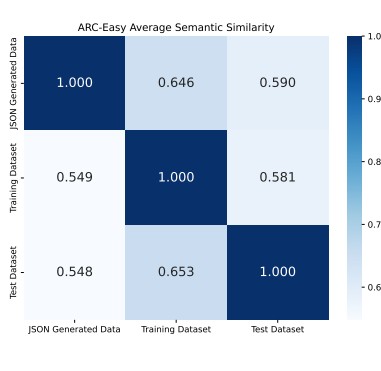 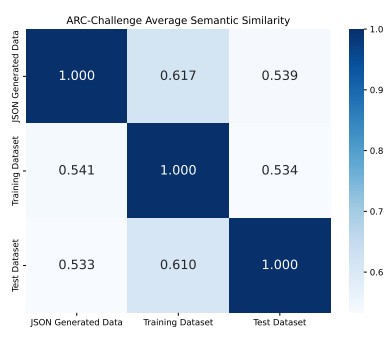

(a) Arc-Easy                    (b) Arc-Challenge

Figure 3: Average Maximum Cosine Similarity between Generated Questions and the Training/Test Sets on ARC-Easy and ARC-Challenge. Similarity is calculated between question embeddings, excluding choices.

### C.8 IS THE RESULTS ONLY FROM LEARNING MCQA FORMAT?

A potential concern is that the performance gains observed with our method might stem solely from learning the structure and format of MCQA, rather than improving actual question-answering ability. To investigate this, we conducted the following cross-evaluation experiment. We used LLaMa-3.1-8B-Instruct to generate 1024 ARC Easy examples using the JSON generation method, along with corresponding LLM-generated scores for distillation. We then trained a DeBERTa-v3-base model on this generated ARC Easy data with distillation, using the same hyperparameters as our main experiments. We compared its performance on MMLU with a model trained directly on MMLU-generated data with distillation and the 5-shot baseline. Results are shown in Table 12.

Training on the ARC Easy-generated data significantly improved performance over the 5-shot baseline. However, the model trained on MMLU-generated data performed significantly better, achieving an average accuracy of 39.3%, compared to 37.9% for the model trained on ARC Easy generated data. This gap suggests that our method is not merely teaching the model the MCQA format, but is also enabling it to acquire task-specific knowledge relevant to the MMLU datasets. Therefore, we conclude that the improvements observed from our method stem from both an improved understanding of the MCQA format and, crucially, an enhanced ability to answer questions within the specific domains covered by MMLU

### C.9 DATASETS SEMANTIC SIMILARITY

To address potential test set contamination, where LLM might have memorized or overfit to the test set during pretraining, and to assess the quality of the generated dataset, we analyzed the semantic similarity between the generated, training, and test set questions using the Sentence Transformers all-MiniLM-L6-v2 model [5]. For each generated question, we calculated the embedding of the question, excluding the choices. Then, we computed the maximum cosine similarity between this embedding and the embeddings of all questions in the training and test sets. We then averaged these maximum similarities across all generated questions to obtain an overall measure of similarity.

Figure 3 shows the average maximum similarity scores. The average maximum similarity between generated questions and the test set is 0.590 for ARC-Easy and 0.539 for ARC-Challenge. These values are comparable to the similarity between the training set and test set (0.581 and 0.534, respectively). If the generated questions were simply copies from the training set, the similarity to the training set would be much higher, and the similarity to the test set would likely also be higher. The observed comparable similarity scores suggest the generated questions are novel and not mere duplicates.

---

[5]https://huggingface.co/sentence-transformers/all-MiniLM-L6-v2

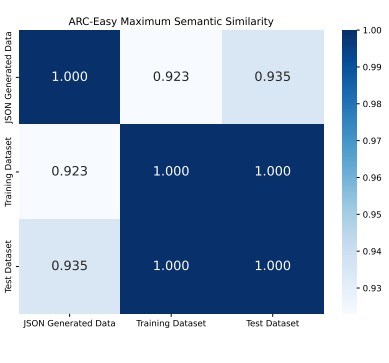 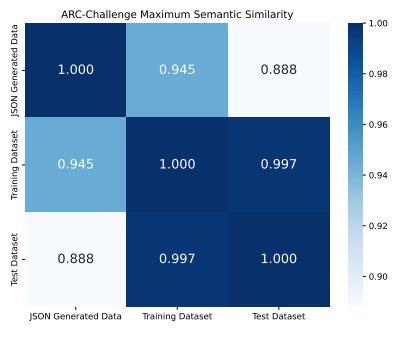

(a) Arc-Easy             (b) Arc-Challenge

Figure 4: Maximum Cosine Similarity Observed between Generated Questions and the Training/Test Sets on ARC-Easy and ARC-Challenge. Similarity is calculated between question embeddings, excluding choices.

To further identify any potential near duplicates, we also examined the maximum similarity scores between the generated questions and the training or test sets. Figure 4 shows these maximum similarity scores. The maximum similarity between the generated data and the test sets is noticeably lower than the maximum similarity between the training and test sets. This further supports our claim that the generated data does not simply replicate the test set questions. The training dataset exhibits near-duplicate questions (similarity near 1), whereas our generated data does not exhibit such high similarity to the test set (around 0.93 and 0.88). The observed semantic similarity between generated and real questions suggests that the LLM is generating questions that are relevant to the target domain and similar in style and complexity to real exam questions. This provides evidence for the quality of the generated data.

### C.10 MMLU DETAILED RESULTS

Tables 13 and 14 present the detailed results for our method on the MMLU benchmark, corresponding to the aggregated results discussed in Section 4.1. Table 13 shows the results for MMLU tasks 0-39, while Table 14 shows the results for tasks 40-56.

## D GENERATED DATASET STATISTIC

This section analyzes the statistical properties of the MMLU datasets generated using both the JSON and decomposed methods. We compare the average and standard deviation of token length in the real, JSON-generated, and decomposed-generated datasets, calculated by concatenating the question and all choices and tokenizing them with the DeBERTa tokenizer. Additionally, we report the parsing success rate for the JSON generation method.

Table 15 presents the statistics of the generated MMLU datasets. Notably, the decomposed method produces data with a significantly higher average token length compared to both the real data and the JSON-generated data. This is likely due to the decomposed method's lack of an inherent filtering mechanism, leading to the generation of more noisy and potentially irrelevant content. For instance, the decomposed method frequently generates excessively long answers, as illustrated in Table 24, where the LLM produced a very long positive answer not typically found in the real data. In contrast, the JSON generation method, by directly mimicking the structure and style of the few-shot examples, tends to generate higher-quality data with lengths closer to the real data. However, despite the increased noise in the decomposed data, decompose generation method surprisingly yields strong performance after applying LLM distillation, as demonstrated in our main experiments.

| | 5-shot | Decomp. generate | Decomp. distill | JSON distill | Tasksource JSON |
|---|---|---|---|---|---|
| Abstract Algebra | 22.4 | 24.2 | 26.0 | 27.2 | 27.6 |
| Anatomy | 24.9 | 34.5 | 36.6 | 34.4 | 41.5 |
| Astronomy | 20.1 | 29.1 | 36.2 | 35.1 | 40.5 |
| Business Ethics | 30.0 | 42.8 | 49.4 | 49.4 | 54.8 |
| Clinical Knowledge | 30.9 | 31.6 | 39.3 | 41.6 | 54.9 |
| College Biology | 23.6 | 33.8 | 36.1 | 36.2 | 42.2 |
| College Chemistry | 27.0 | 25.4 | 29.2 | 26.8 | 30.6 |
| College Computer Science | 36.2 | 26.2 | 32.0 | 33.2 | 37.0 |
| College Mathematics | 27.8 | 22.4 | 22.0 | 24.8 | 27.2 |
| College Medicine | 25.2 | 31.0 | 35.5 | 37.6 | 46.9 |
| College Physics | 25.9 | 16.7 | 23.3 | 25.7 | 30.6 |
| Computer Security | 42.6 | 38.2 | 50.0 | 53.8 | 63.6 |
| Conceptual Physics | 25.4 | 35.2 | 35.3 | 34.3 | 38.5 |
| Econometrics | 25.6 | 24.2 | 23.7 | 21.6 | 27.0 |
| Electrical Engineering | 27.7 | 27.3 | 32.4 | 38.2 | 44.3 |
| Elementary Mathematics | 24.1 | 24.9 | 28.3 | 25.7 | 30.4 |
| Formal Logic | 28.7 | 33.2 | 23.5 | 24.6 | 26.8 |
| Global Facts | 20.4 | 23.4 | 30.4 | 26.4 | 26.8 |
| High School Biology | 23.7 | 38.8 | 42.1 | 40.2 | 51.6 |
| High School Chemistry | 28.8 | 24.0 | 26.5 | 30.0 | 31.2 |
| High School Computer Science | 24.8 | 28.6 | 34.2 | 35.4 | 49.8 |
| High School European History | 25.6 | 39.3 | 49.2 | 50.8 | 66.1 |
| High School Geography | 26.8 | 44.7 | 48.6 | 48.1 | 66.8 |
| High School Government And Politics | 32.3 | 45.2 | 52.0 | 54.0 | 71.9 |
| High School Macroeconomics | 26.8 | 32.2 | 42.1 | 39.9 | 50.9 |
| High School Mathematics | 29.8 | 15.2 | 25.6 | 26.5 | 27.3 |
| High School Microeconomics | 28.3 | 27.9 | 36.5 | 34.0 | 50.3 |
| High School Physics | 25.0 | 25.0 | 26.8 | 28.3 | 27.2 |
| High School Psychology | 33.9 | 39.3 | 48.3 | 51.1 | 67.7 |
| High School Statistics | 28.1 | 27.1 | 30.4 | 30.6 | 35.2 |
| High School Us History | 24.8 | 40.8 | 50.4 | 45.5 | 59.6 |
| High School World History | 33.7 | 46.4 | 53.1 | 51.5 | 70.1 |
| Human Aging | 24.8 | 31.7 | 33.6 | 30.6 | 48.9 |
| Human Sexuality | 32.2 | 38.2 | 40.8 | 46.6 | 55.7 |
| International Law | 44.8 | 29.3 | 56.4 | 65.6 | 66.0 |
| Jurisprudence | 24.3 | 30.2 | 44.6 | 49.3 | 64.4 |
| Logical Fallacies | 29.4 | 46.1 | 51.5 | 55.5 | 63.7 |
| Machine Learning | 26.6 | 27.1 | 28.4 | 30.4 | 29.8 |
| Management | 32.0 | 40.8 | 47.2 | 51.1 | 64.5 |
| Marketing | 43.8 | 49.1 | 60.0 | 61.7 | 77.4 |

Table 13: Result on MMLU[:40] datasets.

| | 5-shot | Decomp. generate | Decomp. distill | JSON distill | Tasksource JSON |
|---|---|---|---|---|---|
| Medical Genetics | 30.6 | 42.8 | 40.8 | 41.0 | 42.0 |
| Miscellaneous | 37.8 | 50.6 | 52.7 | 53.1 | 66.7 |
| Moral Disputes | 28.4 | 28.0 | 39.3 | 41.8 | 55.5 |
| Moral Scenarios | 24.3 | 24.2 | 24.1 | 24.4 | 33.0 |
| Nutrition | 25.6 | 33.9 | 38.1 | 41.8 | 53.1 |
| Philosophy | 30.9 | 40.1 | 42.4 | 43.2 | 54.8 |
| Prehistory | 26.4 | 34.9 | 40.6 | 42.4 | 53.8 |
| Professional Accounting | 24.2 | 25.0 | 29.2 | 28.7 | 36.5 |
| Professional Law | 24.1 | 27.9 | 31.0 | 30.5 | 33.0 |
| Professional Medicine | 26.0 | 29.8 | 34.6 | 28.7 | 39.3 |
| Professional Psychology | 22.9 | 31.3 | 34.7 | 35.3 | 45.8 |
| Public Relations | 26.9 | 41.1 | 44.7 | 42.0 | 57.6 |
| Security Studies | 31.8 | 40.2 | 37.8 | 41.3 | 50.9 |
| Sociology | 30.0 | 33.4 | 48.0 | 50.3 | 65.1 |
| Us Foreign Policy | 40.4 | 42.8 | 52.0 | 53.8 | 65.6 |
| Virology | 24.7 | 28.7 | 32.8 | 36.1 | 39.8 |
| World Religions | 42.3 | 38.9 | 47.8 | 50.8 | 56.1 |
| Humanities | 29.8 | 35.3 | 42.6 | 44.3 | 54.1 |
| STEM | 27.1 | 27.6 | 31.6 | 32.5 | 37.2 |
| Social Science | 29.8 | 36.7 | 42.4 | 43.2 | 56.3 |
| Others | 28.9 | 35.5 | 40.3 | 40.6 | 50.1 |
| All Average | 28.7 | 33.1 | 38.4 | 39.3 | 48.0 |

Table 14: Result on MMLU[40:] datasets.

# E    PROMPT LIST

This section provides the prompts used for data generation and scoring in our experiments. Tables 17 and 18 show the JSON generation prompts and the 5-shot examples for ARC-Easy and ARC-Challenge, respectively. For the decomposed generation method, Table 19 presents the question generation prompt, Table 20 shows the positive answer generation prompt, and Table 21 illustrates the negative answer generation prompt. For paraphrase baseline, the prompt is shown in Table 23. Finally, Table 22 provides an example of the prompt used for scoring the choices with the LLM.

# F    GENERATION EXAMPLE

This section provides examples of the MCQA data generated by our proposed methods. We also note on Decompose generation method, the answer will always be the first choice, as it is the first one to be generated. Our training method with DeBERTa is agnostic to choice permutation, thus using same label over all training data will not have any impact on student model training. Table 24 illustrates a case where the decomposed method generated a noisy positive answer with an excessively long sequence. Table 25 shows an example where the JSON generation method produced an incorrect label, but the LLM scoring was able to identify the correct answer. We also include few generation example on MMLU datasets, which is shown in Table 26 to Table 30

| Dataset Name | Real Length | Decompose Length | JSON length | JSON Parseable |
|---|---|---|---|---|
| Abstract Algebra | $62 \pm 19$ | $198 \pm 141$ | $62 \pm 22$ | 12.7 |
| Anatomy | $57 \pm 19$ | $93 \pm 45$ | $60 \pm 19$ | 42.9 |
| Astronomy | $70 \pm 26$ | $173 \pm 57$ | $104 \pm 45$ | 40.5 |
| Business Ethics | $79 \pm 31$ | $127 \pm 46$ | $108 \pm 38$ | 39.6 |
| Clinical Knowledge | $57 \pm 16$ | $120 \pm 70$ | $69 \pm 24$ | 49.5 |
| College Biology | $72 \pm 35$ | $107 \pm 52$ | $78 \pm 29$ | 43.1 |
| College Chemistry | $72 \pm 33$ | $199 \pm 113$ | $85 \pm 32$ | 20.0 |
| College Computer Science | $105 \pm 47$ | $155 \pm 49$ | $106 \pm 44$ | 13.4 |
| College Mathematics | $77 \pm 30$ | $216 \pm 101$ | $94 \pm 33$ | 4.9 |
| College Medicine | $102 \pm 151$ | $134 \pm 73$ | $75 \pm 29$ | 52.7 |
| College Physics | $76 \pm 20$ | $151 \pm 80$ | $81 \pm 27$ | 27.6 |
| Computer Security | $66 \pm 40$ | $78 \pm 31$ | $65 \pm 27$ | 49.8 |
| Conceptual Physics | $43 \pm 12$ | $81 \pm 38$ | $62 \pm 22$ | 56.0 |
| Econometrics | $98 \pm 38$ | $157 \pm 74$ | $94 \pm 36$ | 30.4 |
| Electrical Engineering | $44 \pm 11$ | $118 \pm 75$ | $65 \pm 19$ | 39.4 |
| Elementary Mathematics | $56 \pm 22$ | $214 \pm 110$ | $69 \pm 29$ | 22.7 |
| Formal Logic | $107 \pm 43$ | $308 \pm 108$ | $92 \pm 36$ | 5.2 |
| Global Facts | $49 \pm 17$ | $80 \pm 44$ | $55 \pm 18$ | 44.0 |
| High School Biology | $74 \pm 31$ | $99 \pm 59$ | $62 \pm 22$ | 51.7 |
| High School Chemistry | $77 \pm 35$ | $177 \pm 109$ | $65 \pm 24$ | 32.7 |
| High School Computer Science | $106 \pm 62$ | $127 \pm 47$ | $80 \pm 35$ | 34.5 |
| High School European History | $334 \pm 117$ | $457 \pm 184$ | $267 \pm 157$ | 13.6 |
| High School Geography | $47 \pm 15$ | $83 \pm 36$ | $58 \pm 22$ | 54.1 |
| High School Government And Politics | $68 \pm 21$ | $116 \pm 48$ | $72 \pm 27$ | 50.8 |
| High School Macroeconomics | $63 \pm 18$ | $102 \pm 41$ | $68 \pm 25$ | 48.0 |
| High School Mathematics | $70 \pm 28$ | $205 \pm 124$ | $77 \pm 29$ | 10.8 |
| High School Microeconomics | $70 \pm 24$ | $102 \pm 43$ | $78 \pm 29$ | 44.6 |
| High School Physics | $95 \pm 38$ | $189 \pm 112$ | $80 \pm 25$ | 19.5 |
| High School Psychology | $61 \pm 31$ | $116 \pm 56$ | $75 \pm 28$ | 48.6 |
| High School Statistics | $115 \pm 42$ | $181 \pm 60$ | $110 \pm 47$ | 17.5 |
| High School Us History | $296 \pm 71$ | $382 \pm 171$ | $256 \pm 144$ | 12.7 |
| High School World History | $332 \pm 126$ | $262 \pm 144$ | $134 \pm 81$ | 17.7 |
| Human Aging | $46 \pm 13$ | $73 \pm 29$ | $59 \pm 18$ | 45.4 |
| Human Sexuality | $55 \pm 24$ | $83 \pm 38$ | $64 \pm 23$ | 52.8 |
| International Law | $86 \pm 23$ | $222 \pm 52$ | $116 \pm 42$ | 39.2 |
| Jurisprudence | $68 \pm 25$ | $91 \pm 46$ | $62 \pm 20$ | 54.2 |
| Logical Fallacies | $66 \pm 28$ | $88 \pm 33$ | $66 \pm 23$ | 45.1 |
| Machine Learning | $77 \pm 39$ | $153 \pm 65$ | $95 \pm 43$ | 41.3 |
| Management | $42 \pm 10$ | $67 \pm 32$ | $55 \pm 18$ | 48.4 |
| Marketing | $60 \pm 16$ | $87 \pm 36$ | $67 \pm 23$ | 58.4 |

Table 15: Statistics of MMLU[:40] datasets.

| Dataset Name | Real Length | Decompose Length | JSON length | JSON Parseable |
|---|---|---|---|---|
| Medical Genetics | 51 ± 13 | 83 ± 41 | 63 ± 23 | 56.3 |
| Miscellaneous | 45 ± 25 | 60 ± 35 | 39 ± 11 | 56.9 |
| Moral Disputes | 67 ± 22 | 158 ± 68 | 97 ± 40 | 45.6 |
| Moral Scenarios | 101 ± 7 | 198 ± 92 | 126 ± 47 | 11.3 |
| Nutrition | 64 ± 26 | 135 ± 63 | 92 ± 39 | 38.5 |
| Philosophy | 60 ± 29 | 78 ± 32 | 56 ± 17 | 55.3 |
| Prehistory | 63 ± 24 | 130 ± 44 | 92 ± 32 | 33.7 |
| Professional Accounting | 96 ± 30 | 172 ± 83 | 103 ± 41 | 25.9 |
| Professional Law | 249 ± 95 | 424 ± 184 | 214 ± 108 | 22.5 |
| Professional Medicine | 169 ± 62 | 232 ± 148 | 139 ± 52 | 22.1 |
| Professional Psychology | 75 ± 32 | 123 ± 59 | 85 ± 36 | 44.5 |
| Public Relations | 56 ± 29 | 98 ± 47 | 66 ± 23 | 45.6 |
| Security Studies | 152 ± 70 | 290 ± 81 | 203 ± 100 | 35.4 |
| Sociology | 66 ± 19 | 92 ± 40 | 72 ± 27 | 49.0 |
| Us Foreign Policy | 59 ± 15 | 140 ± 49 | 74 ± 26 | 41.0 |
| Virology | 52 ± 26 | 83 ± 32 | 69 ± 24 | 52.8 |
| World Religions | 38 ± 8 | 69 ± 33 | 46 ± 14 | 42.8 |

Table 16: Statistic of MMLU[40:] datasets.

---

**System**: You are a bot that excel at creating question about the given topics, and will create question in JSON format!
*##### Few-Shot Example 1:*
**User**: create a question about grade school science!
**Assistant**: {'question': 'Which of the following materials would best slow the transfer of heat?', 'choices': ['aluminum', 'copper', 'glass', 'wood'], 'answer': 3}
*##### Few-Shot Example 2:*
**User**: create a question about grade school science!
**Assistant**: {'question': 'In which environment is white fur color an advantage for survival?', 'choices': ['desert', 'grassland', 'arctic tundra', 'temperate forest'], 'answer': 2}
*##### Few-Shot Example 3:*
**User**: create a question about grade school science!
**Assistant**:{'question': 'The mathematical model for calculating speed is shown below. Speed = distance/time. An airplane traveled 700 kilometers in two hours during a trip. What was the average speed of the plane during the trip?', 'choices': ['5.8 kilometers per hour', '350 kilometers per hour', '1400 kilometers per hour', '84,000 kilometers per hour'], 'answer': 1}
*##### Few-Shot Example 4:*
**User**: create a question about grade school science!
**Assistant**:{'question': 'The aloe plant can absorb a lot of water during a rain shower. The extra water is stored in its leaves. The ability to store water in its leaves is most likely an adaptation to which type of environment?', 'choices': ['one near the ocean', 'one with dry conditions', 'one with a variety of organisms', 'one that receives a lot of sunlight'], 'answer': 1}
*##### Few-Shot Example 5:*
**User**: create a question about grade school science!
**Assistant**:{'question': "Near Earth's equator, water generally exists naturally in the liquid and gas states. In which other part of Earth is water usually found naturally in only two states?", 'choices': ['Indian Ocean', 'interior of Africa', 'South Pole', 'Tropic of Cancer'], 'answer': 2}

**User**: create a question about grade school science!
**Assistant**:

Table 17: Examples of the MCQA data generation prompt using JSON method and the five data used as few-shot example on ARC-Easy.

---

**System**: You are a bot that excel at creating question about the given topics, and will create question in JSON format!

*##### Few-Shot Example 1:*

**User**: create a question about grade school science!

**Assistant**: {'question': 'Which characteristic can a human offspring inherit?', 'choices': ['facial scar', 'blue eyes', 'long hair', 'broken leg'], 'answer': 1}

*##### Few-Shot Example 2:*

**User**: create a question about grade school science!

**Assistant**: {'question': 'Many horses grow a heavy coat in autumn and shed that coat in spring. Scientists were not sure if temperature or the amount of daylight per day (called the photoperiod) caused the change. So, they conducted an experiment and concluded that the change in photoperiod was responsible for the biological changes. Shedding caused by which set of conditions would have helped them arrive at this conclusion?', 'choices': ['a constant photoperiod, but varied temperatures', 'a varied photoperiod, and varied temperatures', 'a constant photoperiod, and a constant temperature', 'a varied photoperiod, but a constant temperature'], 'answer': 3}

*##### Few-Shot Example 3:*

**User**: create a question about grade school science!

**Assistant**:{'question': 'Which is evidence of a chemical reaction?', 'choices': ['the light produced by magnesium when burned', 'the evaporation of water from a solution', 'the fizzing of a soft drink', 'the heat from a light bulb'], 'answer': 0}

*##### Few-Shot Example 4:*

**User**: create a question about grade school science!

**Assistant**:{'question': 'A class is studying the density of rock samples. What scientific equipment do they need to determine the density of the rock samples?', 'choices': ['microscope and balance', 'beaker and graduated cylinder', 'graduated cylinder and balance', 'microscope and graduated cylinder'], 'answer': 2}

*##### Few-Shot Example 5:*

**User**: create a question about grade school science!

**Assistant**:{'question': 'Which energy resource is non-renewable?', 'choices': ['oil', 'solar', 'water', 'wind'], 'answer': 0}

**User**: create a question about grade school science!

**Assistant**:

---

Table 18: Examples of the MCQA data generation prompt using JSON method and the five data used as few-shot example on ARC-Challenge.

---

**System**: You are a bot that excel at creating question about the given topics!

**User**: create a question about grade school science!

**Assistant**: Which of the following materials would best slow the transfer of heat?

**User**: create a question about grade school science!

**Assistant**: The mathematical model for calculating speed is shown below. Speed = distance/time. An airplane traveled 700 kilometers in two hours during a trip. What was the average speed of the plane during the trip?

(Other 5 shot example)

**User**: create a question about grade school science!

**Assistant**:

---

Table 19: Examples of the question generation prompt.

**System**: You are a bot that excel at answering question and will answer all question correctly(answer shortly)!
**User**: Which of the following materials would best slow the transfer of heat?
**Assistant**: wood
**User**: The mathematical model for calculating speed is shown below. Speed = distance/time. An airplane traveled 700 kilometers in two hours during a trip. What was the average speed of the plane during the trip?
**Assistant**: 350 kilometers per hour

(Other 5 shot example)

**User**: The aloe plant can absorb a lot of water during a rain shower. The extra water is stored in its leaves. The ability to store water in its leaves is most likely an adaptation to which type of environment?
**Assistant**:

Table 20: Examples of the positive answer generation prompt.

**System**: You are a bot that always answer question with possible but wrong answer and reply with diverse answer(answer shortly)!
**User**: Answer the question with wrong but possible answer and use different answer from the Forbidden Answer!
Question: Which of the following materials would best slow the transfer of heat?
Forbidden Answer :
- wood
- copper
Answer:
**Assistant**: aluminum
**User**: Answer the question with wrong but possible answer and use different answer from the Forbidden Answer!
Question: The mathematical model for calculating speed is shown below. Speed = distance/time. An airplane traveled 700 kilometers in two hours during a trip. What was the average speed of the plane during the trip?
Forbidden Answer :
- 350 kilometers per hour
- 1400 kilometers per hour
- 5.8 kilometers per hour Answer:
**Assistant**: 84,000 kilometers per hour

(Other 5 shot example)

**User**: Answer the question with wrong but possible answer and use different answer from the Forbidden Answer!
Question: The aloe plant can absorb a lot of water during a rain shower. The extra water is stored in its leaves. The ability to store water in its leaves is most likely an adaptation to which type of environment?
Forbidden Answer :
- one with dry conditions
Answer:
**Assistant**:

Table 21: Examples of the negative answer generation prompt.

**User**: Which of the following materials would best slow the transfer of heat?
A. aluminum
B. copper
C. glass
D. wood
**Assistant**: D
**User**: The mathematical model for calculating speed is shown below. Speed = distance/time. An airplane traveled 700 kilometers in two hours during a trip. What was the average speed of the plane during the trip?
A. 5.8 kilometers per hour
B. 350 kilometers per hour
C. 1400 kilometers per hour
D. 84,000 kilometers per hour
**Assistant**: B

(Other 5 shot example)

**User**: The aloe plant can absorb a lot of water during a rain shower. The extra water is stored in its leaves. The ability to store water in its leaves is most likely an adaptation to which type of environment?
A. one near the ocean
B. one with dry conditions
C. one with a variety of organisms
D. one that receives a lot of sunlight
**Assistant**:

Table 22: Examples of the LLMs MCQA scoring prompt.

**System**: You are a bot that excel at paraphrasing.
*##### Few-Shot Example 1:*
**User**: paraphrase this : AI is transforming various sectors by taking over tasks that used to require human labor. While this advancement can lead to greater efficiency and cost savings, it also sparks worries about job loss and the moral considerations surrounding AI-driven choices.
**Assistant**: Artificial intelligence is revolutionizing industries by automating tasks that were previously performed by humans. This technology has the potential to increase efficiency and reduce costs, but it also raises concerns about job displacement and the ethical implications of AI decision-making.
*##### Few-Shot Example 2:*
**User**: paraphrase this : Despite the challenges they faced during the project, the team managed to deliver a high-quality product that exceeded the client's expectations.
**Assistant**: Even with the obstacles encountered throughout the project, the team successfully produced a top-notch product that went beyond what the client had anticipated.
*##### Few-Shot Example 3:*
**User**: paraphrase this : Happy
**Assistant**: Joyful
*##### Few-Shot Example 4:*
**User**: paraphrase this : What are the main factors that contribute to climate change, and how do they each affect the environment?
**Assistant**: What key elements drive climate change, and what impact does each one have on the environment?
*##### Few-Shot Example 5:*
**User**: paraphrase this : The project lead is Sarah.
**Assistant**: Sarah is leading the project.

Table 23: Few-shot prompt for paraphrase baselines.

What is 6 basic physical changes but isn't considered chemical is happening? Example of these include, Melting Ice changing shape into a water molecules without making atoms get mixed up or recombine, also making a ball of dough gets compressed
Choice 1) Those examples are of physical changes. 6 examples are:
1. Melting
2. Boiling
3. Freezing
4. Compressing
5. Changing shape (or compressing) a ball of dough
6. Evaporation
Choice 2) Dissolving sugar in water
Choice 3) Sublimation
Choice 4) evaporation
Choice 5) Changing the color of a white shirt.
Choice 6) melting a metal statue in a fire.
Answer : 1

Table 24: example of Decompose Generation with positive choice containing long sequences instead of short answer.

Plants make a sweet tasty treat in large organs called ______ inside their stems.
A. fruits
B. seeds
C. roots
D. leave
Initial Answer : B
LLM Probability Score : [(A) 35.1%, (B) 21.4%, (C) 21.3%, (D) 22.2%]

Table 25: example of Wrong label when generating data directly with JSON method and how distillation could helps.

To categorize a viral reemergercy does it need specific molecular features such as sequence of a certain nucleocapsid, structure of its envelope, specific replication methods and what one or a different option.
A. Yes, including serologic cross-reactivity with other members of the same virus.
B. The presence of a tail of variable length
C. The virus being of aquatic origin
D. Mutual seroneutralization with another reemerging virus
E. The presence of a peculiarly patterned nucleic acid methylation
F. The virus being of terrestrial origin
Initial Answer : A
LLM Probability Score : [(A) 33.4%, (B) 12.3%, (C) 7.8%, (D) 22.2%, (E) 17.9%, (F) 6.4%]

Table 26: Generated data example using Decompose generation method with MMLU dataset Virology.

Loss of which bodily function is most directly attributed to the gradual decrease in dopamine receptors associated with aging?
A. Motivation
B. Regulation of body temperature
C. Regulation of appetite
D. Coordination
E. Memory
F. Regulation of sleep
Initial Answer : A
LLM Probability Score : [(A) 28.4%, (B) 8.7%, (C) 15.8%, (D) 14.7%, (E) 16.0%, (F) 16.3%]

Table 27: Generated data example using Decompose generation method with MMLU dataset Human Aging.

---

This question refers to the following information.

We may imagine, if we please, that all white inhabitants of this Province (for, at present, the inhabitants do neither read nor talk but for white People). that these white inhabitants were all the owners in their Own right as to goods (money goods) except so few that we do not want and those but a Hand ful they having lost all and taken this Course to beg and Stealing: which is as clear that I believe even from all Accounts as that some have and will go farther than to Stealers which is as great as the devil would for one to make himself King of Virginia... They have some hopes some way or an oother to get that Land on the Sea side. And yet they all Conceived a Jealousi[e] to take the best Part, especially about this Town and River. in that part so far we have kept clear their Town and as to them Land all those who were from this year from North England but there were and was the most averse than the rest... The greatest Body went out of the River... to which this place has yet seen, but of which one and twenty of this Colony have fallen in.

The first written passage about early American Settlement, is attributed to:

A. Captain John Smith
B. William Bradford
C. John Rolfe
D. John Winthrop
E. Christopher Newport
F. William Penn
Initial Answer : A
LLM Probability Score : [(A) 23.8%, (B) 15.5%, (C) 18.5%, (D) 18.2%, (E) 11.7%, (F) 12.3%]

---

Table 28: Generated data example using Decompose generation method with MMLU dataset High School US History.

---

As per studies, which vitamin deficiency, linked to malnutrition in aged patients is commonly reported
A. Vitamin A deficiency
B. Vitamin C Deficiency
C. Vitamin D Deficiency
D. Biotin Deficiency
Initial Answer : C
LLM Probability Score : [(A) 7.5%, (B) 14.5%, (C) 69.6%, (D) 8.5%]

---

Table 29: Generated data example using JSON generation method with MMLU dataset Human Aging.

---

According to UK nutritional reference intakes (RNI), what amount of water for adults aged over 16, considering a temperature of 22- 27°C was stated (as of 2020)?
A. At least 30 mL. day/ per day for a normal inactive woman's diet
B. On average 2ltr water per person per year
C. 75 mL.day /per person per dayfor an active healthy adult diet
D. Less than none
Initial Answer : B
LLM Probability Score : [(A) 29.7%, (B) 13.3%, (C) 49.0%, (D) 8.1%]

---

Table 30: Generated data example using JSON generation method with MMLU dataset Nutrition.

