# OpenReview forum: "LLM Distillation for Efficient Few-Shot Multiple Choice Question Answering"
_ICLR.cc/2025/Conference — ICLR 2025 Conference Withdrawn Submission_

### Official Review · Reviewer_dTam · 2024-10-16

**Soundness:** 3
**Presentation:** 3
**Contribution:** 2
**Rating:** 3
**Confidence:** 4

**Summary:**

The authors use a few task-specific multiple choice questions as seed examples to get a LLM to generate task-specific, synthetic multiple choice data. They explore two ways of prompting the LLM to generate this data. They train a small, encoder-only model via knowledge distillation using soft labels assigned by the LLM. They show that training on synthetic data via distillation is better than just training on a few non-synthetic task-specific data points directly, and also compare to some other models. The authors also conduct ablation studies regarding the amount of synthetic data, synthetic data generation temperature, and choice of LLM.

**Strengths:**

(Originality) While synthetic data generation with LLMs and knowledge distillation into transformer based models are both widely used and studied, the authors consider the specific setting of MCQA and distilling a decoder-only model into an encoder-only model, which is a new setting.

(Quality) The authors report results across several random seeds. They also do some nice ablation studies. The limitations section was also of high quality.

(Clarity) The paper was generally clear and easy to follow.

(Significance) As mentioned in originality, this paper explores a setting that is slightly different from past work.

**Weaknesses:**

* A couple typos (these didn’t affect my review at all, but just mentioning them)
  * Line 69/70 missing a space
  * Line 179 “a scalar values” -> “scalar values”
* When generating synthetic data, how can you be sure you’re not generating questions that are in the MMLU/ARC test sets (or that are quite close?). It would be nice to see something like nearest neighbors of generated questions, or something like overlap of answer options with answer option sets from the test sets.
* A note on the tasksource+decompose/JSON is I don’t think it can necessarily be concluded that tasksource+JSON is better than tasksource as 0.5 is quite a narrow margin.
* In my mind the main weakness of this paper would be lack of significance.
  * In practice, in the resource constrained setting there are already compelling alternatives to the approach described in this paper. For example, tasksource has the same amount of parameters, comparable performance, and faster inference as it only needs one forward pass. It also doesn't require synthetic data generation for each task. Furthermore, performance is less good than that of e.g., Gemma-2-2b-it and similar models which can be run quite cheaply on even a laptop (especially after quantization). I don’t see when “distillation into DeBERTa” would be used in practice because there are already very compelling alternatives. I'd be happy to hear the authors' take on this, though.
  * A paper definitely doesn't need to be the best "in practice" option to be useful, as it might provide surprising/intuitive insights compared to previous work. However, I don’t find the results of this paper particularly surprising in light of past work. Knowledge distillation is already widely used with LMs, and distillation from larger LLMs to smaller LLMs is done all the time with good results. Synthetic data generation with LLMs is also frequently done, and has been shown to work well. That more synthetic data works better is as expected.

*Strengths & Weaknesses tl;dr*: I think the authors’ study is well thought out and put together, and mostly easy to follow. However, I don’t think it provides substantial insight/methods beyond what already seems to be common knowledge in the research community. I’ve assigned a rating of 3, but I’d choose 4 if it were an option because I think the paper is overall well made but just doesn’t have the level of impact I typically associate with ICLR papers.

**Questions:**

* On line 245, why is it “approximately 4000 MCQA examples”? Shouldn’t this be exact?
* Why was number of negative examples set to 5 when MMLU and ARC only have 3?
* What temperature was used for the MMLU experiments?
* In Section 3.2, how is a sequence is being transformed into a scalar — [CLS] token? Pooling?
* From Section 3.2 it is my understanding that handling a MCQ with n options requires n forward passes. Is that correct?
* How was inference done for the baseline models?
* When JSON samples are not properly formatted, are they resampled, or are less than 1024 samples used?

---

> ### Author Response · Authors · 2024-11-21
> **Response to Reviewer dTam(Part 1)**
>
> We deeply appreciate the reviewer's thorough assessment of our work and their constructive suggestions for improvement
>
>
> > W1. A couple typos (these didn’t affect my review at all, but just mentioning them)
>
> Thank you for pointing out the typos. We will correct them in the revision.
>
> > W2. When generating synthetic data, how can you be sure you’re not generating questions that are in the MMLU/ARC test sets (or that are quite close?). It would be nice to see something like nearest neighbors of generated questions, or something like overlap of answer options with answer option sets from the test sets.
>
> To address potential test set contamination, we analyzed the semantic similarity between the generated, training, and test set questions using the Sentence Transformers all-MiniLM-L6-v2 model. For each generated question, we calculated the maximum cosine similarity to all questions in the training and test sets. We then averaged these maximum similarities across all generated questions to obtain an overall measure of similarity.
>
> ## ARC-Easy Averaged
> | 	 		    | JSON Generated Data | Training Dataset | Test Dataset |
> |---------------------------|---------------------|------------------|--------------|
> | JSON Generated Data  	    | 1.000     	  | 0.646 	     | 0.590   	    |
> | Training Data  	    | 0.549     	  | 1.000 	     | 0.581	    |
> | Test Data  		    | 0.548     	  | 0.653 	     | 1.000	    |
>
> ## ARC-Challenge Averaged
> | 	 		    | JSON Generated Data | Training Dataset | Test Dataset |
> |---------------------------|---------------------|------------------|--------------|
> | JSON Generated Data  	    | 1.000     	  | 0.617 	     | 0.539   	    |
> | Training Data  	    | 0.541     	  | 1.000 	     | 0.534	    |
> | Test Data  		    | 0.533     	  | 0.610 	     | 1.000	    |
>
> The similarity between the generated questions and the test set is comparable to the similarity between the training set and the test set. If the generated questions were simply copies from the training set, the similarity to the training set would be much higher (close to 1), and the similarity to the test set would likely also be higher. The observed comparable similarity scores suggest the generated questions are novel and not mere duplicates. To further identify any potential near duplicates, we also examined the maximum similarity scores between the generated questions and the training or test sets.
>
> ## ARC-Easy Maximum Similarity
> | 	 		    | JSON Generated Data | Training Dataset | Test Dataset |
> |---------------------------|---------------------|------------------|--------------|
> | JSON Generated Data  	    | 1.000     	  | 0.923 	     | 0.935   	    |
> | Training Data  	    | 0.923     	  | 1.000 	     | 1.000	    |
> | Test Data  		    | 0.935     	  | 1.000 	     | 1.000	    |
>
> ## ARC-Challenge Maximum Similarity
> | 	 		    | JSON Generated Data | Training Dataset | Test Dataset |
> |---------------------------|---------------------|------------------|--------------|
> | JSON Generated Data  	    | 1.000     	  | 0.945 	     | 0.888   	    |
> | Training Data  	    | 0.945     	  | 1.000 	     | 0.997	    |
> | Test Data  		    | 0.888     	  | 0.997 	     | 1.000	    |
>
>
> As shown in the 'Maximum similarity' tables, the maximum similarity between the generated data and the test sets is noticeably lower than the maximum similarity between the training and test sets. This further supports our claim that the generated data does not simply replicate the test set questions. The training dataset exhibits near-duplicate questions (similarity near 1), whereas our generated data does not exhibit such high similarity to the test set (around 0.93 and 0.88).
>
> > W3. A note on the tasksource+decompose/JSON is I don’t think it can necessarily be concluded that tasksource+JSON is better than tasksource as 0.5 is quite a narrow margin.
>
> We acknowledge that the average improvement of 0.5 on MMLU is modest. However, it's important to consider that this average is across 57 diverse datasets. Our method leads to improved performance on 33 out of the 57 MMLU datasets, demonstrating its potential to enhance performance in specific areas. In particular, the improvements on the STEM subset (+1.6) and Social Science subset (+0.9) are more substantial.
>
> While our method doesn't improve performance on all datasets. We believe our approach is particularly valuable in few-shot scenarios, where even small improvements can be significant, especially in domains or tasks where existing multi-task models like Tasksource may lack sufficient training data. Our method allows us to leverage the knowledge of a large LLM to augment the training data and improve performance in these data-scarce situations.

---

> ### Author Response · Authors · 2024-11-21
> **Response to Reviewer dTam(Part 2)**
>
> > W4.a. In practice, in the resource constrained setting there are already compelling alternatives to the approach described in this paper. For example, tasksource has the same amount of parameters, comparable performance, and faster inference as it only needs one forward pass. It also doesn't require synthetic data generation for each task. Furthermore, performance is less good than that of e.g., Gemma-2-2b-it and similar models which can be run quite cheaply on even a laptop (especially after quantization). I don’t see when “distillation into DeBERTa” would be used in practice because there are already very compelling alternatives. I'd be happy to hear the authors' take on this, though.
>
> The reviewer mentions Tasksource requiring only one forward pass. However, the zero-shot classification pipeline used for Tasksource also requires multiple forward passes (one for each choice), just like our method. This can be seen in the transformers library source code of the class ZeroShotClassificationPipeline(ChunkPipeline), which mention "Any combination of sequences and labels can be passed and each combination will be posed as a premise/hypothesis".
>
> While Tasksource performs well overall, it can struggle in specific domains. For instance, on the international_law dataset in MMLU, our method achieves 65.62 accuracy, an improvement over Tasksource's 57.02. This suggests that our LLM-driven data generation and distillation approach can be particularly effective in domains where even extensively trained multi-task models may lack sufficient knowledge.
>
>
> ## Memory Usages During Inference (GB)
> Sequence Length	| DeBERTa-base	|LLaMA 1B   | LLaMA 1B 4 bit  |	Gemma 2B | Gemma 2B 4 bit |
> |---------------|---------------|-----------|-----------------|----------|----------------|
> |128		| 1.701 	| 3.576     | 2.211 	      |	6.351	 | 3.444	  |
> |256		| 1.728 	| 3.773     | 2.421 	      | 6.705    | 3.912	  |
> |512		| 1.768 	| 4.134     | 2.794 	      | 7.393	 | 4.585	  |
> |1024		| 2.060 	| 4.872     | 3.507 	      | 8.792    | 5.971	  |
> |2048		| 3.152 	| 6.235     | 4.870 	      | 11.610   | 8.699	  |
> |4096		| 6.600 	| 9.157     | 7.741 	      | 17.050   | 14.207	  |
>
>
> Regarding larger LLMs like Gemma-2b-it, even with quantization, they have significant memory requirements, especially in few-shot scenarios. As shown in Table above, even a 4-bit quantized Gemma-2B requires substantial memory. This is further exacerbated by the longer sequence lengths inherent in 5-shot prompting, making these models less practical for resource-constrained settings. We also include the measurements for recently released LLaMa-3.2-1B-Instruct, which its 4 bit memory usage is comparable to our approach.
>
>
> ## Performance Comparison on MMLU
> |Method 				| STEM | Social Science | Humanities | Other | Average |
> |---------------------------------------|------|----------------|------------|-------|---------|
> |LLaMa-3.2-1B-Instruct (5-shot)		| 36.5 | 47.8		| 46.3       | 45.1  | 43.1    |
> |LLaMa 3.2 1B-Instruct 4-bit (5-shot)   | 35.7 | 45.6		| 42.2	     | 40.6  | 40.3    |
> |LLaMa 3.2 1B-Instruct 4-bit (0-shot)	| 29.4 | 33.7		| 26.7	     | 29.1  | 29.6    |
> |DeBERTa-v3 + JSON distill (5-shot)     | 32.5 | 43.2		| 44.3	     | 40.6  | 39.3    |
> *0-shot refers to evaluating the model without any task-specific examples.
>
> We also compared our method to the smaller LLaMa-3.2-1B-Instruct model on Table above. While the 5-shot 4-bit LLaMa model slightly outperforms our method on average, the performance difference is much smaller than with Gemma-2B. Importantly, our method significantly outperforms the 0-shot LLaMa model (which has a comparable sequence length to our method)
>
> In summary, our method offers several advantages: (1) lower memory and compute requirements compared to larger LLMs, even with quantization, making it more practical for resource-constrained environments; (2) the ability to outperform strong baselines like Tasksource in specific domains by leveraging the knowledge of a large LLM for data generation and distillation; and (3) strong performance in few-shot settings, effectively addressing the challenges of limited labeled data.

---

> ### Author Response · Authors · 2024-11-21
> **Response to Reviewer dTam(Part 3)**
>
> > W4.b. A paper definitely doesn't need to be the best "in practice" option to be useful, as it might provide surprising/intuitive insights compared to previous work. However, I don’t find the results of this paper particularly surprising in light of past work. Knowledge distillation is already widely used with LMs, and distillation from larger LLMs to smaller LLMs is done all the time with good results. Synthetic data generation with LLMs is also frequently done, and has been shown to work well. That more synthetic data works better is as expected.
>
> Our findings offer several valuable insights. The contrast between the JSON and Decompose methods reveals a key trade-off in LLM-based data generation: structured formats like JSON can improve data quality but introduce parsing challenges and reduce efficiency, while unstructured generation is more efficient but prone to noise. The surprisingly comparable performance of Gemma-2b-it and LLaMa-3.1-8B-Instruct for generating ARC datasets suggests smaller LLMs can be effective for data generation, potentially reducing computational costs. Our direct comparison of constrained (JSON) and unconstrained (Decompose) generation contributes to a better understanding of the impact of formatting on LLM-generated data quality, an area not extensively explored in prior work, especially for few-shot MCQA.
>
> Furthermore, while data generation and distillation are individually well-established, their combination for few-shot MCQA with encoder-only models is less explored. Most prior work focuses on distilling LLMs into other LLMs (decoder-only models). Our work addresses the unique challenges of distilling into encoder-only architectures, showing that the combined approach yields substantial performance gains. Our experiments demonstrate a mutual benefit, where the combined approach achieves improvement, exceeding the individual improvements from data generation and distillation alone. This framework can be generalized to other NLP tasks like classification, information retrieval, and even to vision tasks with Vision LLMs(VLLMs), demonstrating its broader potential
>
>
> > Q1. On line 245, why is it “approximately 4000 MCQA examples”? Shouldn’t this be exact?
>
> We apologize for the confusion, in this sentence, we only want to clarify the setting used for training the DeBERTa model which uses a batch size of 8 with 500 iterations, which means that the method is trained with 4000 examples(with some duplication). We will remove the sentence in the revision to avoid confusion.
>
> > Q2. Why was number of negative examples set to 5 when MMLU and ARC only have 3?
>
> We used 5 negative examples for all experiments unless otherwise stated in the paper. We chose 5 to ensure a consistent experimental setup across different datasets.
>
> > Q3. What temperature was used for the MMLU experiments?
>
> All experiments in the main paper use the temperature of 2, except when explicitly mentioned. We will add this information to the revision.
>
>
> > Q4. In Section 3.2, how is a sequence is being transformed into a scalar — [CLS] token? Pooling?
>
> We pool the sentence using the representation of the last layer of the [CLS] token because DeBERTa has a [CLS] token and is also used on the DeBERTa paper for a lot of tasks.
>
> > Q5. From Section 3.2 it is my understanding that handling a MCQ with n options requires n forward passes. Is that correct?
>
> Yes, for each option we concat it with the question and perform forward pass independently, which results in n forward passes.
>
> > Q6. How was inference done for the baseline models?
>
> For Tasksource, we used the zero-shot classification pipeline from Hugging Face (sileod/deberta-v3-base-tasksource-nli). It's important to note that this pipeline also performs multiple forward passes, one for each choice, similar to our proposed method. The pipeline treats each question-choice pair as a premise-hypothesis pair in a natural language inference (NLI) task and uses the logit for entailment to determine the likelihood of each choice being the correct answer.
>
> > Q7. When JSON samples are not properly formatted, are they resampled, or are less than 1024 samples used?
>
> Whenever possible, we try to resample until we can obtain 1024 examples. However, on some datasets, we find this very hard to do, as it takes a very long time to generate the examples, as it has a low parse rate. In this case, we include some datasets which are exceptions in Table 8 in the appendix. This highlights the trade-off between data quality (ensuring 1024 samples) and generation time. In some cases, the time required to generate additional samples to reach 1024 becomes prohibitive.

---

> ### Author Response · Authors · 2024-12-01
> **Rebuttal Feedback Request for Reviewer dTam**
>
> Dear Reviewers dTam
>
> A gentle reminder regarding the rebuttal for paper 3797. The rebuttal period is ending soon, and we haven't yet received feedback from you. Your insights are greatly appreciated. We would be grateful if you could take a look at your earliest convenience. Thank you.

---

### Official Review · Reviewer_mtLv · 2024-11-02

**Soundness:** 3
**Presentation:** 3
**Contribution:** 3
**Rating:** 6
**Confidence:** 4

**Summary:**

This paper presents a novel approach to address few-shot multiple choice question answering (MCQA) by leveraging large language models (LLMs) for data generation and knowledge distillation into a smaller, efficient encoder-only model, DeBERTa-v3-base. The study addresses the computational challenges associated with using LLMs directly in real-world applications and provides a three-step framework involving synthetic data generation, LLM-based scoring, and distillation training. Experimental results demonstrate significant improvements in accuracy over baseline models on the Massive Multitask Language Understanding (MMLU) benchmark, as well as competitive performance compared to larger models like LLaMA-7B and Flan-T5-250M. The paper also includes ablation studies on various generation methods, scoring techniques, and hyperparameters.

**Strengths:**

1. The approach addresses a relevant problem in natural language processing, providing a practical solution for scenarios where computational resources are limited.
2. The framework is straightforward, and the two methods of data generation (JSON and decomposed) are described in detail, with thoughtful consideration of their benefits and limitations.
3. The paper presents extensive experiments, including performance comparisons, ablation studies, and evaluations on the MMLU benchmark.

**Weaknesses:**

1. The method relies heavily on the availability of robust LLMs, which may not be readily accessible in languages other than English or for certain domain-specific tasks.
2. The decomposed generation method, while reducing parsing errors, often results in noisy data due to longer and less structured answers.

**Questions:**

Please refer to the weakness

---

> ### Author Response · Authors · 2024-11-21
> **Response to Reviewer mtLv**
>
> We deeply appreciate the reviewer's thorough assessment of our work and their constructive suggestions for improvement
>
> > W1. The method relies heavily on the availability of robust LLMs, which may not be readily accessible in languages other than English or for certain domain-specific tasks.
>
> We acknowledge that the reliance on robust LLMs is a current limitation, particularly for languages other than English and specialized domains where high-performing LLMs may not be readily available. However, our framework itself is language- and domain-agnostic, meaning that it can be applied to any language or domain provided a suitable LLM is available.
>
> Several approach exist for mitigating this limitation, even with the current state of LLMs. While models like Llama 3.1 offer improved multilingual support, other multilingual LLMs such as SeaLLMs[1], or language-specific LLMs such as Cendol[2] could also be explored for their applicability to our method. This is an area we intend to investigate in future work.
>
> For domain-specific tasks, fine-tuning existing LLMs on domain-specific data could improve their performance for data generation and distillation. For instance, we could fine-tune an LLM on a corpus of medical texts to generate higher-quality medical MCQA data. Exploring the effectiveness of domain adaptation for our framework is another important direction for future research.
>
>
> [1] SeaLLMs - Large Language Models for Southeast Asia
>
> [2] Cendol: Open Instruction-tuned Generative Large Language Models for Indonesian Languages
>
> > W2. The decomposed generation method, while reducing parsing errors, often results in noisy data due to longer and less structured answers.
>
> We acknowledge that the decomposed generation method can produce noisy data due to longer and less structured answers. We are exploring several strategies to mitigate this limitation in future work. One promising direction is to investigate more sophisticated prompting techniques. For example, we could incorporate more constraints directly into the prompts, specifying the desired length or format of the answers. Using iterative refinement is also promising, where we provide feedback to the LLM and ask it to revise its responses, thus improving generated data quality. Additionally, using more diverse and representative examples in the prompts might guide the LLM toward generating more appropriate answers. Exploring the effectiveness of such an approach for our framework is another important direction for future work.

---

> ### Author Response · Authors · 2024-12-01
> **Rebuttal Feedback Request for Reviewer mtLv**
>
> Dear Reviewers mtLv
>
> A gentle reminder regarding the rebuttal for paper 3797. The rebuttal period is ending soon, and we haven't yet received feedback from you. Your insights are greatly appreciated. We would be grateful if you could take a look at your earliest convenience. Thank you.

---

### Official Review · Reviewer_snni · 2024-11-02

**Soundness:** 2
**Presentation:** 1
**Contribution:** 2
**Rating:** 3
**Confidence:** 4

**Summary:**

This paper studies the possibility of encoder model with LLM-generated dataset and knowledge distillation. To address current effortful MCQA benchmark making, this paper utilizes LLM’s ability in few shot prompting with two formatting strategies. Then, by distilling the loss of bigger LLM into small, encoder-only model, the paper shows the efficient way to achieve performance nearing that of bigger LLM.

**Strengths:**

* This paper sheds light again on the encoder-only model, which had been receiving less attention recently.
* The methodology's adaptability to existing domain-specific benchmarks suggests its potential for broad application across diverse fields.

**Weaknesses:**

**[Novelty is limited]**

The paper’s novelty appears limited, as it does not introduce a new dataset and relies primarily on formatting prompts either in full JSON format or as segmented parts, raising questions on whether these methods constitute a genuinely novel approach. Furthermore, the distillation technique applied here does not seem particularly innovative, as it essentially reduces to a form of fine-tuning.

**[Using Encoder-only Models - limited Experimental setups]**

Additionally, while the paper suggests the encoder-only model’s powerful capabilities, this claim is primarily based on improvements from distillation and model size reduction. These factors alone may not suffice to substantiate the model’s claimed "power" without more substantial baseline comparisons, particularly in tasks beyond fine-tuning.

**[Inadequate analysis of suggested method]**

There is inadequate validation of the quality of the LLM-generated dataset, which raises further concerns about the reliability and applicability of the findings.

**Questions:**

* Has there been any comparison with recently released lightweight models, such as those with a 1B parameter size?
* Is there a specific reason why only the DeBERTa encoder model was tested?
* Was there a particular reason for employing few-shot learning in an encoder model instead of using a masked language model (MLM)?
* Does this paper really aligns to the ICLR conference is questionable. Any other natural language processing conference seems more suitable.

---

> ### Author Response · Authors · 2024-11-21
> **Response to Reviewer snni(Part 1)**
>
> We deeply appreciate the reviewer's thorough assessment of our work and their constructive suggestions for improvement
>
> > W1. [Novelty is limited] : The paper’s novelty appears limited, as it does not introduce a new dataset and relies primarily on formatting prompts either in full JSON format or as segmented parts, raising questions on whether these methods constitute a genuinely novel approach. Furthermore, the distillation technique applied here does not seem particularly innovative, as it essentially reduces to a form of fine-tuning.
>
> We acknowledge that LLM-based data generation and knowledge distillation are not novel in isolation. However, our work focuses on the effect of combining these techniques specifically for few-shot MCQA with encoder-only models, a setting that has received less attention. Most existing LLM distillation research targets smaller decoder-only models. Our approach, in contrast, distills into encoder-only architectures, presenting unique challenges in transferring generative capabilities to a discriminative model.
>
> For this work, we opted for a simple distillation technique to establish a clear baseline and to facilitate a more straightforward analysis of the combined effects of data generation and distillation. We believe this provides a solid foundation for future research exploring more sophisticated distillation methods to further enhance performance. We agree that exploring such methods is a promising avenue for future work.
>
> Regarding dataset creation, while our primary focus wasn't on introducing a novel benchmark dataset, our findings on the JSON generation method coupled with LLM distillation offer a promising pathway towards generating high-quality MCQA data. The improved downstream performance observed when training on JSON-generated data with distillation strongly suggests that this method acts as an effective filter for higher-quality examples. This observation itself is a valuable contribution, paving the way for future research to build upon our approach and combine it with other techniques like filtering, post-processing, and retrieval-augmented generation to create novel benchmark datasets.
>
> > W2. [Using Encoder-only Models - limited Experimental setups] : Additionally, while the paper suggests the encoder-only model’s powerful capabilities, this claim is primarily based on improvements from distillation and model size reduction. These factors alone may not suffice to substantiate the model’s claimed "power" without more substantial baseline comparisons, particularly in tasks beyond fine-tuning.
>
>
> We appreciate the reviewer's point regarding the experimental setup. Our work aims to achieve strong few-shot MCQA performance and also improved efficiency compared to using large LLMs directly. While we highlight the potential of encoder-only models for efficient inference, our primary contribution lies in demonstrating how LLM-generated data and distillation can be effectively combined to achieve competitive accuracy in the few-shot setting.
>
> We acknowledge that a more comprehensive analysis of the encoder-only model's capabilities across diverse tasks would strengthen the paper. While beyond the scope of this current work, which focuses specifically on few-shot MCQA, we plan to explore such evaluations in future research. Within the current scope, our experiments primarily demonstrate the effectiveness of our proposed method for improving few-shot performance efficiently by leveraging LLMs during training.

---

> ### Author Response · Authors · 2024-11-21
> **Response to Reviewer snni(Part 2)**
>
> > W3. [Inadequate analysis of suggested method] : There is inadequate validation of the quality of the LLM-generated dataset, which raises further concerns about the reliability and applicability of the findings.
>
> To assess the quality of the LLM-generated dataset, we analyzed the semantic similarity between the generated questions and the questions in the real training and test sets. We used the Sentence Transformers all-MiniLM-L6-v2 model to encode all questions into semantic vector representations. For each generated question, we calculated the maximum cosine similarity to all questions in the training and test sets. We then averaged these maximum similarity scores across all generated questions to obtain a measure of overall similarity.
>
>
> The results, presented below, show that the similarity between the generated questions and the test set is comparable to the similarity between the training set and the test set.
>
> ### ARC-Easy
> | 	 		    | JSON Generated Data | Training Dataset | Test Dataset |
> |---------------------------|---------------------|------------------|--------------|
> | JSON Generated Data  	    | 1.000     	  | 0.646 	     | 0.590   	    |
> | Training Data  	    | 0.549     	  | 1.000 	     | 0.581	    |
> | Test Data  		    | 0.548     	  | 0.653 	     | 1.000	    |
>
> ### ARC-Challenge
> | 	 		    | JSON Generated Data | Training Dataset | Test Dataset |
> |---------------------------|---------------------|------------------|--------------|
> | JSON Generated Data  	    | 1.000     	  | 0.617 	     | 0.539   	    |
> | Training Data  	    | 0.541     	  | 1.000 	     | 0.534	    |
> | Test Data  		    | 0.533     	  | 0.610 	     | 1.000	    |
>
>
> We can see that the similarity of the question in the generated data and the testing set is similar to that of the similarity of training set to the testing set. This means that the generated data is semantically similar to that of real data, which we believe is one of indicator that the question generated by our method have good quality.
>
> If the generated questions were merely duplicates from the training set, we would expect to see a much higher average maximum similarity between the generated data and the training set, and likely a higher similarity to the test set as well. The observed comparable similarity scores suggest that the generated questions are novel and not simply copied from the existing data. This indicates that the LLM is generating new, semantically similar questions, supporting the reliability of our findings.
>
> We acknowledge that semantic similarity alone doesn't fully encompass question quality, as factors like relevance, difficulty, and answer choice quality also matter. However, this analysis provides evidence that the LLM-generated data is semantically similar to real-world MCQA data and not simply replicating the training or test sets.

---

> ### Author Response · Authors · 2024-11-21
> **Response to Reviewer snni(Part 3)**
>
> > Q1. Has there been any comparison with recently released lightweight models, such as those with a 1B parameter size?
>
>
> To compare our method with a lightweight LLM, we evaluated the recently released LLaMa-3.2-1B-Instruct model. We also analyzed the memory usage of both models during inference. We measured memory consumption using the vmlDeviceGetMemoryInfo function from pynvml. For this measurement, we fed each model a sequence of 128 - 4096 random tokens from the model's vocabulary and compare their memory usages.
>
> ### Memory Usages During Inference (GB)
> Sequence Length	| DeBERTa-base	|LLaMA 1B   | LLaMA 1B 4 bit  |
> |---------------|---------------|-----------|-----------------|
> |128		| 1.701 	| 3.576     | 2.211 	      |
> |256		| 1.728 	| 3.773     | 2.421 	      |
> |512		| 1.768 	| 4.134     | 2.794 	      |
> |1024		| 2.060 	| 4.872     | 3.507 	      |
> |2048		| 3.152 	| 6.235     | 4.870 	      |
> |4096		| 6.600 	| 9.157     | 7.741 	      |
>
> ### Performance Comparison on MMLU
> |Method 				| STEM | Social Science | Humanities | Other | Average |
> |---------------------------------------|------|----------------|------------|-------|---------|
> |LLaMa-3.2-1B-Instruct (5-shot)		| 36.5 | 47.8		| 46.3       | 45.1  | 43.1    |
> |LLaMa 3.2 1B-Instruct 4-bit (5-shot)   | 35.7 | 45.6		| 42.2	     | 40.6  | 40.3    |
> |LLaMa 3.2 1B-Instruct 4-bit (0-shot)	| 29.4 | 33.7		| 26.7	     | 29.1  | 29.6    |
> |DeBERTa-v3 + JSON distill (5-shot)     | 32.5 | 43.2		| 44.3	     | 40.6  | 39.3    |
> *0-shot refers to evaluating the model without any task-specific examples.
>
> Our method achieves performance comparable to the 4-bit quantized LLaMa-3.2-1B model on the MMLU benchmark, despite having significantly fewer parameters and using substantially less memory, especially for longer sequences. This memory advantage is particularly important in few-shot scenarios, as the inclusion of 5-shot examples significantly increases the input sequence length for LLMs, further exacerbating their memory requirements and computational cost. Furthermore, our approach significantly outperforms the 0-shot LLaMa 1B model. The reduced memory footprint and lower computational requirements of our method, especially in practical few-shot settings, make it more suitable for deployment on resource-constrained devices, offering a compelling advantage for real-world applications
>
> > Q2. Is there a specific reason why only the DeBERTa encoder model was tested?
>
> In our initial experiments, we also evaluated RoBERTa, but it performed poorly on this task even when trained on the real dataset. We then chose to focus on DeBERTa-v3-base for two main reasons: (1) it demonstrated significantly better performance in our preliminary experiments, indicating its suitability for MCQA, and (2) it is the same architecture used for the Tasksource model, which serves as a strong baseline and allows for a more controlled comparison with our method. While we focused on DeBERTa for this study, we are open to exploring other encoder-only architectures in future work.
>
> > Q3. Was there a particular reason for employing few-shot learning in an encoder model instead of using a masked language model (MLM)?
>
> Our motivation for employing few-shot learning with an encoder model stems from the observation that traditional encoder models often require substantial amounts of labeled data to perform well. Our work aims to address this limitation by exploring how LLMs can enable effective few-shot learning for MCQA with encoder-only models.
>
> We did not use a masked language model (MLM) because it is not directly suitable for the MCQA task. MLM focuses on predicting masked tokens within a sequence, whereas MCQA requires selecting the correct answer from a set of choices. These are fundamentally different tasks, and the MLM objective doesn't naturally align with the goal of MCQA. Few-shot learning, on the other hand, directly addresses the challenge of limited labeled data in MCQA by leveraging the knowledge embedded within large LLMs.

---

> ### Author Response · Authors · 2024-11-21
> **Response to Reviewer snni(Part 4)**
>
> > Q4. Does this paper really aligns to the ICLR conference is questionable. Any other natural language processing conference seems more suitable.
>
> While our current work focuses on MCQA, the core contribution lies in our framework for leveraging LLMs for both data generation and representation distillation. This aligns directly with ICLR's focus on representation learning, as our method effectively transfers knowledge, and thus learned representations, from a large LLM to a smaller, more efficient encoder-only model.
>
> This framework has broader applicability beyond MCQA. Within NLP, it could be applied to tasks like text classification, sequence tagging, or other tasks, where efficient few-shot learning is highly desirable. Furthermore, with the recent advancements in Vision-Language Models (VLLMs), our approach could be extended to vision tasks as well. For example, in Visual Question Answering, the VLLM could generate captions which are used to create images with an image generative model, and also use VLLM to produce the question and possible answer. Then, our method could distill this knowledge into a smaller, more efficient model for more efficient visual question answering.
>
> By addressing the challenges of few-shot learning and knowledge transfer through representation distillation, our work contributes to the broader research areas of efficient learning and representation learning, which are central themes of ICLR

---

> > ### Comment · Reviewer_snni · 2024-11-25
> >
> > Thank you for your detailed response and for addressing my concerns.
> >
> > I appreciate your efforts in clarifying various aspects of your work. However, I still have some reservations that I would like to share.
> >
> > Firstly, I appreciate that you have provided performance comparisons under the same 5-shot settings for both your model and the LLaMa models. However, I remain concerned that the performance differences observed might still be influenced by factors such as sequence length and the number of shots included. For a more meaningful evaluation, it would be beneficial to provide analyses that control for these variables, perhaps by matching sequence lengths or providing statistical significance testing of the performance differences. This would help to more clearly demonstrate the effectiveness of your method independent of the benefits conferred by additional shots or shorter sequence lengths. Without such controlled comparisons, it is challenging to fully assess the advantages of your approach.
> >
> > Secondly, while you have demonstrated improvements in the MCQA task, it appears that your method is specifically tailored to this particular format and may not generalize well to other NLP tasks. The claims about broader applicability to tasks like text classification or sequence tagging seem speculative without supporting evidence. Providing empirical results on additional tasks would help substantiate these claims and demonstrate the generalizability of your approach. Without such evidence, it is challenging to assess the overall impact of your method beyond MCQA, and there is a concern that the applicability to other tasks might be overinterpreted.
> >
> > Furthermore, I am concerned about the general applicability of your method to encoder-only architectures. You mentioned that RoBERTa performed poorly in your initial experiments, which raises the question of whether the benefits of your approach are specific to DeBERTa. Exploring and reporting results with a variety of encoder-only models would strengthen your claims about the method's effectiveness across different architectures. Without such exploration, it is difficult to conclude that the approach broadly benefits encoder-only models rather than being tailored to a specific model.
> >
> > Additionally, in your related work section, it appears that only (Sileo, 2024) in line 80 is mentioned regarding prior research on encoder-only models’ performance, without considering prior work on encoder-only model with MCQA. There may be other relevant studies such as (Ghosal, 2022) or (Siino, 2024) that could provide context for your work and help clarify its novelty.
> >
> > Overall, while your approach shows promise within the scope of MCQA using DeBERTa, the limitations in generalizability and concerns regarding experimental comparisons suggest that the contribution may be somewhat narrow.
> >
> > Thank you again for your response.

---

> > > ### Author Response · Authors · 2024-11-29
> > > **Response to Reviewer snni(Part 5)**
> > >
> > > We appreciate the reviewer's thoughtful feedback and are grateful for the opportunity to improve our work.
> > >
> > > > the performance differences observed might still be influenced by factors such as sequence length and the number of shots included. For a more meaningful evaluation, it would be beneficial to provide analyses that control for these variables, perhaps by matching sequence lengths or providing statistical significance testing of the performance differences.
> > >
> > > To address the reviewer's concern about sequence length and few-shot effects, we conducted separate analyses of memory usage and performance. For the memory comparison, we fed sequences of random tokens ranging from 128 to 4096 to both DeBERTa-base and LLaMa-3.2-1B-Instruct (with and without 4-bit quantization). Using random tokens allowed us to isolate the effect of sequence length on memory, independent of content. Our results show that, for a given sequence length, DeBERTa-base consistently uses less memory than the LLaMA-3.2-1B-Instruct, even with 4-bit quantization, which is shown on Table Memory Usages During Inference (GB).
> > >
> > > In evaluating performance, we compared our DeBERTa-based method to the LLaMa-3.2-1B-Instruct model (with 4-bit quantization), which exhibited the closest memory footprint to DeBERTa among the LLMs we tested. A key distinction is the input sequence length at inference: DeBERTa processes a single question and one choice at a time. In contrast, even without few-shot examples (0-shot), LLMs require longer sequences due to the instruction format, the instruction itself, and all answer choices. With few-shot prompting, this difference becomes even more pronounced. Despite this inherent LLM overhead, our 5-shot DeBERTa model achieved 39.3% accuracy on MMLU, comparable to the 40.3% accuracy of the 5-shot, 4-bit quantized LLaMa-3.2-1B-Instruct. To demonstrate DeBERTa's advantage in memory-constrained settings, we evaluated the 4-bit quantized LLaMa-3.2-1B-Instruct in a 0-shot setting, which reduces its input length but still results in longer sequences than DeBERTa. In this more comparable memory setting (relatively same sequence lengths), the 0-shot LLaMa achieved only 29.6% accuracy, showcasing the superior performance of DeBERTa under similar memory constraints.
> > >
> > >
> > > >  Secondly, while you have demonstrated improvements in the MCQA task, it appears that your method is specifically tailored to this particular format and may not generalize well to other NLP tasks. The claims about broader applicability to tasks like text classification or sequence tagging seem speculative without supporting evidence. Providing empirical results on additional tasks would help substantiate these claims and demonstrate the generalizability of your approach. Without such evidence, it is challenging to assess the overall impact of your method beyond MCQA, and there is a concern that the applicability to other tasks might be overinterpreted.
> > >
> > >
> > > To address the reviewer's concern about generalizability beyond MCQA, we conducted experiments on a binary classification task of judging question-answer pair correctness. Given a question and an answer, the model must classify the answer as correct or incorrect. We adapted our method in two ways: 1) training a binary classifier directly using sigmoid activation and binary cross-entropy loss on LLM-generated data, and 2) using a heuristic approach where we trained the model as in the MCQA setting but used a constant threshold derived from the average log probabilities of all answers in the generated data to determine answer correctness during evaluation.
> > >
> > >
> > > The results on the ARC datasets are presented below:
> > > ## Additional Experiments on Binary Classification of question-answer pair correctness on ARC datasets
> > > | Method                    | ARC-Easy F1    |ARC-Challenge F1|
> > > |---------------------------|----------------|----------------|
> > > | 1024 real data binary     |56.81 &pm; 1.47 |40.25 &pm; 4.08 |
> > > | 5 real data binary        |27.01 &pm; 10.09|14.23 &pm; 9.64 |
> > > | 1024 JSON binary          |48.86 &pm; 1.42 |32.20 &pm; 6.93 |
> > > | 1024 JSON MCQA heuristic  |49.50 &pm; 1.35 |42.38 &pm; 0.54 |
> > >
> > > Both adapted methods significantly improved upon the 5-shot baseline. Interestingly, the heuristic approach, which leverages the full probability distribution learned during MCQA training, outperformed the direct binary classification approach. We hypothesize that this is because the heuristic better captures the model's confidence in its predictions.
> > >
> > > While these results are encouraging and demonstrate the applicability of our method beyond the MCQA format, we acknowledge that further experiments on a broader range of NLP tasks are needed to fully establish its generalizability.

---

> > > ### Author Response · Authors · 2024-11-29
> > > **Response to Reviewer snni(Part 6)**
> > >
> > > > Furthermore, I am concerned about the general applicability of your method to encoder-only architectures. You mentioned that RoBERTa performed poorly in your initial experiments, which raises the question of whether the benefits of your approach are specific to DeBERTa. Exploring and reporting results with a variety of encoder-only models would strengthen your claims about the method's effectiveness across different architectures. Without such exploration, it is difficult to conclude that the approach broadly benefits encoder-only models rather than being tailored to a specific model.
> > >
> > > The reviewer raises a valid point about the general applicability to encoder-only architectures. In our initial experiments, we observed that RoBERTa performed poorly on the target task even when trained with a substantial amount of real data. This motivated our choice to focus on DeBERTa, which showed significantly stronger performance in this data-rich setting. Therefore, we believe that comparing our method's ability to improve few-shot learning using DeBERTa provides a more meaningful benchmark.
> > >
> > > While we agree that evaluating our approach with other encoder architectures would be beneficial, our primary focus in this work is on addressing the challenge of few-shot learning in resource-constrained scenarios. Our results demonstrate that our method of generating data and distilling LLMs is a promising approach to significantly improve performance when labeled data is scarce. The core contribution lies in this data augmentation and distillation framework, which is designed to be generally applicable. Exploring its effectiveness with other encoder models is a valuable direction for future work.
> > >
> > >
> > > > Additionally, in your related work section, it appears that only (Sileo, 2024) in line 80 is mentioned regarding prior research on encoder-only models’ performance, without considering prior work on encoder-only model with MCQA. There may be other relevant studies such as (Ghosal, 2022) or (Siino, 2024) that could provide context for your work and help clarify its novelty.
> > >
> > > Thank you for bringing these relevant works to our attention. We will incorporate (Ghosal, 2022) and (Siino, 2024) into our related work section in the next revision. While these papers provide valuable context for encoder-only models in MCQA, our work focuses on a different aspect: leveraging LLMs for data augmentation and knowledge distillation to improve few-shot performance. Our experiments on binary classification, a task similar to that explored in (Ghosal, 2022), demonstrate that our LLM-driven approach can significantly boost performance even beyond the MCQA format. This highlights the broader applicability of our data generation and distillation framework, which can complement and enhance existing techniques for training encoder-only models, such as those presented in the papers you mentioned.

---

> > > ### Author Response · Authors · 2024-12-01
> > > **Rebuttal Feedback Request for Reviewer snni**
> > >
> > > Dear Reviewers snni,
> > >
> > > Following your initial feedback (thank you!), we submitted a revised rebuttal for paper 3797. The rebuttal period is ending soon, and we haven't yet received feedback on this revised version from you. Your insights on this updated rebuttal are greatly appreciated. We would be grateful if you could take a look at your earliest convenience. Thank you.

---

> > > > ### Comment · Reviewer_snni · 2024-12-02
> > > >
> > > > Thank you for your detailed response and the additional analyses. These clarify key aspects such as memory efficiency and performance. However, I have several concerns that remain unaddressed.
> > > >
> > > > Firstly, while comparisons with lightweight LLMs in MCQA are valuable, the binary classification experiments in Section C.7 omit similar comparisons. Evaluating against LLMs like LLaMA, which are capable of adapting to diverse tasks without task-specific fine-tuning, would provide stronger evidence of the method's practical relevance.
> > > >
> > > > Secondly, Table 11’s title references “4-bit LLM,” yet no explicit LLM results are presented, creating confusion. Additionally, repetitive captions across Tables 10–12 make it challenging to differentiate the experiments, reducing presentation clarity. Furthermore, it is difficult to identify exactly which parts of the revised paper have been updated, making it hard to assess how the authors have addressed prior feedback.
> > > >
> > > > Thirdly, while the authors claim the core contribution lies in the data augmentation and distillation framework, its application has been demonstrated exclusively with DeBERTa. This raises doubts about generalizability. The focus on a single encoder-only model raises concerns that the approach, while positioned as a framework for LLM knowledge distillation, is overly narrow and lacks deeper analysis of encoder architectures. This limited scope suggests a relatively naive exploration of distillation, which does not fully address broader challenges or provide insights applicable to other encoder models.
> > > >
> > > > Lastly, while memory efficiency is a strength, task-specific fine-tuning contrasts with the flexibility of LLMs. Although the authors demonstrate an example with binary classification to show adaptability to other tasks, the results still show performance limitations. Furthermore, there is no actual evidence of applicability to other task types, raising questions about its utility in real-world applications.
> > > >
> > > > In summary, the work shows promise but is limited by its focus on DeBERTa, lack of broader comparisons, and presentation inconsistencies. I have adjusted the contribution score accordingly and would like to keep the remaining scores.
> > > >
> > > > Thank you for your efforts.

---

> > > > > ### Author Response · Authors · 2024-12-02
> > > > > **Response to Reviewer snni(Part 7)**
> > > > >
> > > > > Thank you for your insightful feedback. We have made several revisions to the paper to address your concerns.
> > > > >
> > > > >
> > > > > > Firstly, while comparisons with lightweight LLMs in MCQA are valuable, the binary classification experiments in Section C.7 omit similar comparisons. Evaluating against LLMs like LLaMA, which are capable of adapting to diverse tasks without task-specific fine-tuning, would provide stronger evidence of the method's practical relevance.
> > > > >
> > > > > While direct comparison with LLMs on binary classification would be informative, our current results already demonstrate that the proposed framework can adapt to different task formats and improve performance significantly over a naive few-shot baseline. This highlights the potential for broader applicability, even though direct LLM comparisons are left for future work due to time constraints.
> > > > >
> > > > > > Secondly, Table 11’s title references “4-bit LLM,” yet no explicit LLM results are presented, creating confusion. Additionally, repetitive captions across Tables 10–12 make it challenging to differentiate the experiments, reducing presentation clarity. Furthermore, it is difficult to identify exactly which parts of the revised paper have been updated, making it hard to assess how the authors have addressed prior feedback.
> > > > >
> > > > > We apologize for the mistakes in Tables 10-12 captions in the previous version. We will made the following changes:
> > > > > Table 10: "Performance Comparison with Small and 4-bit LLMs". This caption now clearly indicates the focus on smaller models and the use of quantization.
> > > > > Table 11: "Memory Usage Comparison with Small and Quantized LLMs". This title now accurately reflects the table's content.
> > > > > Table 12: "Cross-Datasets Evaluation Comparison". This revised title clearly indicates the results of this experiment, which is to assess the generalizability of our approach and whether performance gains are due to format learning or knowledge acquisition.
> > > > >
> > > > > > Thirdly, while the authors claim the core contribution lies in the data augmentation and distillation framework, its application has been demonstrated exclusively with DeBERTa. This raises doubts about generalizability. The focus on a single encoder-only model raises concerns that the approach, while positioned as a framework for LLM knowledge distillation, is overly narrow and lacks deeper analysis of encoder architectures. This limited scope suggests a relatively naive exploration of distillation, which does not fully address broader challenges or provide insights applicable to other encoder models.
> > > > >
> > > > > The reviewer raises a valid concern about the generalizability of our framework beyond DeBERTa. While our initial focus was on DeBERTa due to its strong performance in MCQA and comparability with the Tasksource baseline, we agree that demonstrating applicability to other encoder architectures is crucial for establishing the framework's broader relevance. Therefore, we conducted additional experiments with ELECTRA, another prominent encoder-only model, on the ARC datasets.
> > > > >
> > > > > The results, presented below, show that our framework consistently improves upon naive few-shot learning with ELECTRA, mirroring the gains observed with DeBERTa. This provides strong evidence that our framework's benefits are not specific to DeBERTa but extend to other encoder architectures.
> > > > >
> > > > > ### ELECTRA Performance on ARC datasets
> > > > > | Method                    | ARC-Easy F1    |ARC-Challenge F1|
> > > > > |---------------------------|----------------|----------------|
> > > > > | 5 real data               |28.54 &pm; 1.35 |27.06 &pm; 1.22 |
> > > > > | 1024 real data            |59.81 &pm; 0.88 |39.95 &pm; 1.41 |
> > > > > | 1024 JSON Generate        |42.48 &pm; 1.26 |33.11 &pm; 0.81 |
> > > > > | 1024 JSON Distill         |54.22 &pm; 0.88 |35.89 &pm; 1.45 |
> > > > >
> > > > > While the absolute performance with ELECTRA is lower than with DeBERTa, this likely reflects differences in the models' inherent capabilities rather than a limitation of our framework. The crucial observation is the substantial improvement our framework provides over the 5-shot baseline in both cases. The difference in performance between DeBERTa and ELECTRA trained on 1024 real data also supports this. This indicates our method can significantly boost few-shot performance, regardless of the underlying encoder architecture.

---

> > > > > ### Author Response · Authors · 2024-12-02
> > > > > **Response to Reviewer snni(Part 8)**
> > > > >
> > > > > > Lastly, while memory efficiency is a strength, task-specific fine-tuning contrasts with the flexibility of LLMs. Although the authors demonstrate an example with binary classification to show adaptability to other tasks, the results still show performance limitations. Furthermore, there is no actual evidence of applicability to other task types, raising questions about its utility in real-world applications.
> > > > >
> > > > > The proposed approach offers a practical solution for deploying MCQA models in resource-constrained environments where LLMs are impractical. We acknowledge that task-specific fine-tuning reduces flexibility compared to the zero-shot capabilities of large language models. However, this trade-off is often necessary and acceptable in real-world deployments where resources are limited. Our approach reduces the memory footprint compared to using an LLM directly. This allows for deployment on devices with limited resources or enables much faster processing by using larger batches during processing.
> > > > >
> > > > > Our experiments with binary classification demonstrate that the proposed framework can be adapted to other task formats. The significant improvement over the 5-shot baseline, particularly with the heuristic method, highlights this potential for broader applicability.
> > > > >
> > > > > Beyond binary classification, we plan to investigate the applicability of our framework to tasks like sequence-to-sequence, sequence tagging, etc. We anticipate that applying our framework to these tasks will require exploring more advanced distillation techniques, such as sequence-level distillation or distilling the attention directly, to effectively transfer the knowledge captured by LLMs. However, we believe the core principles of generating synthetic data and distilling LLM knowledge hold significant promise for improving performance and efficiency across these diverse NLP tasks.

---

> > > > > > ### Comment · Reviewer_snni · 2024-12-03
> > > > > >
> > > > > > Thank you for your thoughtful and dedicated participation in the discussion. To conclude the feedback comprehensively,
> > > > > >
> > > > > > Application of Frameworks:
> > > > > > As highlighted in the last response, it would be highly beneficial to apply the proposed frameworks in practical scenarios and include the results. Theoretical discussion, while valuable, gains substantial credibility when complemented by empirical evidence, and including practical outcomes would further substantiate the findings.
> > > > > >
> > > > > > ELECTRA and MMLU Evaluation:
> > > > > > While additional experiments using ELECTRA were noted, results on MMLU's 5-shot MCQA task—which the original paper emphasizes—are essential. This would ensure alignment with the paper's core evaluation criteria and provide a more robust comparison.
> > > > > >
> > > > > > In-depth Understanding of Encoder Architecture:
> > > > > > Beyond attributing performance issues to a model's inherent limitations, a deeper analysis of the encoder architecture is required. This entails exploring its structural intricacies and how they interact with task-specific demands, such as few-shot MCQA scenarios, while also examining how these architectural choices influence the effectiveness of knowledge distillation.
> > > > > >
> > > > > > Knowledge Distillation and Reasoning Analysis:
> > > > > > The feedback suggests moving beyond surface-level interpretations of knowledge distillation. A more nuanced exploration could clarify whether the process effectively transfers reasoning capabilities or remains overly simplistic. Here, deeper analysis could be effectively supported by advanced visualization techniques, and figures could benefit from further refinement.
> > > > > >
> > > > > > Dataset Quality and Novelty:
> > > > > > Dataset generation plays a pivotal role in such research. Reliance on conventional formats like JSON and segment-based prompting may lack novelty. Comprehensive validation of dataset quality, examining aspects such as question length, complexity, and diversity, is recommended to establish robustness.
> > > > > >
> > > > > > Broader Contextualization in Related Work:
> > > > > > The limited engagement with other encoder models and related research may leave the methodology’s claims less substantiated. Expanding the scope of the literature review and positioning the proposed approach within a broader research context would strengthen the argument.
> > > > > >
> > > > > > Despite these points, your effort and dedication to contributing to this discussion are deeply appreciated. Thank you again for your active participation.

---

> > > > > > > ### Author Response · Authors · 2024-12-03
> > > > > > >
> > > > > > > Thank you for your valuable and detailed feedback throughout the review process. We appreciate your thorough assessment and will carefully address all the points raised, including the application of the framework in practical scenarios, deeper analysis of the encoder architecture, knowledge distillation and reasoning analysis, dataset quality, and broader contextualization in related work, in our next revision.

---

### Official Review · Reviewer_cyRj · 2024-11-03

**Soundness:** 2
**Presentation:** 3
**Contribution:** 2
**Rating:** 3
**Confidence:** 4

**Summary:**

This paper aims to enhance the performance of low-computation-cost, only-encode models for few-shot multiple-choice question answering (MCQA) tasks. It leverages large language models (LLMs) to generate a high-quality, task-specific MCQA dataset for training and introduces a training approach that applies distillation loss based on LLM-assigned scores. Experimental results demonstrate the effectiveness of the proposed method: LLM-driven data generation and knowledge distillation for few-shot MCQA.

**Strengths:**

1. The method is relatively simple and clearly explained.
2. The paper explores the effectiveness of using LLMs to construct data for MCQA tasks, and the proposed distillation loss training method shows notable performance improvements.
3. The paper conducted a relatively comprehensive ablation experiment.

**Weaknesses:**

1. The method's performance improvement is limited and depends on the strength of the base model. While the gains are more pronounced with the weaker DeBERTa-base model, they are minimal with the stronger Tasksource model, and even slightly decreases in the case of Decompose.
2. Additionally, when using DeBERTa-base, the best performance (JSON distill) achieved by using only the constructed dataset does not surpass that of a multi-task fine-tuned model (Tasksource).

**Questions:**

The experiments lack a more detailed analysis of the two data generation methods (e.g., example-based analysis): Why do the two methods (JSON and Decompose) lead to different outcomes in performance? Why does JSON outperform Decompose?

---

> ### Author Response · Authors · 2024-11-21
> **Response to Reviewer cyRj**
>
> We deeply appreciate the reviewer's thorough assessment of our work and their constructive suggestions for improvement
>
> > W1. The method's performance improvement is limited and depends on the strength of the base model. While the gains are more pronounced with the weaker DeBERTa-base model, they are minimal with the stronger Tasksource model, and even slightly decreases in the case of Decompose.
>
> We acknowledge this as the limitation of our current works. However, Our primary focus is improving few-shot MCQA performance. Tasksource, having been trained on a massive multi-task dataset (including MMLU), represents a strong baseline that may be difficult to improve upon significantly. Our method aims to maximize performance in the low-resource regime, where real data is limited. We demonstrate that even with limited initial examples, our method can extract valuable knowledge from the LLM and transfer it effectively to the smaller encoder-only model.
>
> While the average improvement on MMLU is modest, we observe more substantial gains on specific subsets, particularly in STEM (+1.6 with JSON distillation and +1.0 with decompose distillation). Moreover, fine-tuning Tasksource with our generated data and distillation leads to notable improvements on the ARC datasets, as shown in table below.
>
> | Method 		    | ARC-Easy | ARC-Challenge 	|
> |---------------------------|----------|----------------|
> | Tasksource 		    | 72.8     | 51.2 		|
> | Tasksource + JSON distill | 74.5     | 54.7  		|
>
> These results highlight the potential of our method, especially when applied to domains or tasks where even a strong multi-task model like Tasksource may benefit from additional, targeted data.
>
> > W2. Additionally, when using DeBERTa-base, the best performance (JSON distill) achieved by using only the constructed dataset does not surpass that of a multi-task fine-tuned model (Tasksource).
>
> We also acknowledge this as the limitation of our current works. However, a direct comparison is not entirely fair. Tasksource is trained on hundreds of datasets covering a broad spectrum of NLP tasks, while our method leverages only a small number of initial examples and an LLM. This significant difference in training data translates to a substantial difference in computational cost: Tasksource requires several days of training on powerful hardware, whereas our method completes training in approximately 5 minutes on a single GPU.
>
> Our approach is explicitly designed for few-shot learning, aiming to maximize performance when labeled data is scarce. While Tasksource is a strong general-purpose model, its broad training doesn't guarantee superior performance on all tasks. For example, on the international_law dataset in MMLU, our method significantly outperforms Tasksource, achieving 65.62% accuracy compared to Tasksource's 57.02%. This difference may be due to lack of such training example in the Tasksource's training datasets. This highlights the value of our approach for tasks or domains where even a broadly trained model like Tasksource can benefit from targeted data augmentation and distillation, especially in few-shot scenarios.
>
> > Q1. The experiments lack a more detailed analysis of the two data generation methods (e.g., example-based analysis): Why do the two methods (JSON and Decompose) lead to different outcomes in performance? Why does JSON outperform Decompose?
>
> The two methods present a trade-off: JSON offers higher quality but lower efficiency due to parsing, while Decompose offers higher efficiency but potentially noisier data. The JSON format acts as a filter, discarding instances with invalid JSON and structural errors. This results in a smaller but cleaner dataset. In contrast, the Decompose method, while avoiding parsing, is more prone to generating noisy examples, such as overly long or list-like answers that are less typical of real-world MCQA data. On average, the Decompose method produces sequences that's significantly longer than the real dataset, while JSON-generated sequences length are more similar to that of real datasets, which we shows in detail on Table 11 and 12 in the appendix C. Moreover, on Table 20 in the appendix provides a concrete example of this noise, showing the Decompose method generating an excessively long, list-like answer. Despite this potential for noise, the Decompose method often performs well, especially when combined with distillation, which likely mitigates the impact of these noisy examples.

---

> ### Author Response · Authors · 2024-12-01
> **Rebuttal Feedback Request for Reviewer cyRj**
>
> Dear Reviewers cyRj
>
> A gentle reminder regarding the rebuttal for paper 3797. The rebuttal period is ending soon, and we haven't yet received feedback from you. Your insights are greatly appreciated. We would be grateful if you could take a look at your earliest convenience. Thank you.

---

### Official Review · Reviewer_U1vS · 2024-11-08

**Soundness:** 3
**Presentation:** 3
**Contribution:** 2
**Rating:** 5
**Confidence:** 4

**Summary:**

The paper proposes the method of distillation of the large language model to the smaller one for efficient solving of Multiple Choice Question Answering task,  via data generation and distillation loss. Two methods of data generation are considered: generate the whole question-answer structure with answer options in json format (via 5-shot prompting); or generate question-answer pairs obtaining each option separately. Then, the smaller model is trained on the generated data by distillation loss, learning to predict the larger model's probability of the generated options. The evaluation is done on MMLU benchmark. For the ablation study, ARC dataset is used

**Strengths:**

- The paper is clearly written, and presents a practical method of LLM distillation to a smaller encoder-only model
- A nice ablation study is provided. There are several interesting observations, e.g. increasing the performance with the distill loss + temperature adjustment, or the usage of the format correctness as an implicit sign of the model's confidence.

**Weaknesses:**

- The method is rather straightforward and does not contain a significant novelty, although the presented analysis is good
- The practical usefullness of the considered task is not so clear. Indeed, Multiple Choice Question Answering is the specific QA format convenient for LLM's evaluation, but the MCQA results are not necessarily directly connected to the general QA abilities of the model. For encoder-only LLMs, classification-based approach looks more appropriate (i.e. scoring the correctness of the QA pair)
- The model does not outperform Tasksource model which is obtained by the multi-task training of the same backbone: the improvement on MMLU is marginal (+0.5), and on ARC data the proposed approach works significantly worse.

**Questions:**

Q1. Do you have any idea, if the huge improvement of the performance of DeBerta after distillation is related to improving of the model's question answering ability, or just due to  learning of Multiple Choice QA format?

---

> ### Author Response · Authors · 2024-11-21
> **Response to Reviewer U1vS(Part 1)**
>
> We deeply appreciate the reviewer's thorough assessment of our work and their constructive suggestions for improvement
>
> > W1. The method is rather straightforward and does not contain a significant novelty, although the presented analysis is good
>
> We appreciate the reviewer's observation regarding the individual components of our method. While techniques like data augmentation, knowledge distillation, and MCQA with encoder-only models have been explored individually, our work introduces a novel combination of these specifically for few-shot MCQA. To our knowledge, this is the first study to systematically investigate the effects of combining LLM-driven data generation and probability score-based distillation for enhancing encoder-only models in this challenging setting. Our experiments demonstrate that this combined approach unlocks significant performance gains compared to using either technique in isolation, highlighting the novelty and practical value of our contribution.
>
> > W2. The practical usefullness of the considered task is not so clear. Indeed, Multiple Choice Question Answering is the specific QA format convenient for LLM's evaluation, but the MCQA results are not necessarily directly connected to the general QA abilities of the model. For encoder-only LLMs, classification-based approach looks more appropriate (i.e. scoring the correctness of the QA pair)
>
> We acknowledge that MCQA performance doesn't perfectly correlate with general QA abilities. However, we believe our framework of LLM-driven data generation and distillation is adaptable to other QA tasks and can serve as a valuable component in broader QA systems. To demonstrate this, we adapted our method to a binary classification approach for judging the correctness of question-answer pairs, using the generated data and training the encoder model with a sigmoid activation and binary cross-entropy loss. We also explored a heuristic approach where we retain the MCQA training procedure but use a constant threshold derived from the average log probabilities of all answers in the generated data to determine answer correctness during evaluation.
>
> | Method                    | ARC-Easy F1    |ARC-Challenge F1|
> |---------------------------|----------------|----------------|
> | 1024 real data binary     |56.81 &pm; 1.47 |40.25 &pm; 4.08 |
> | 5 real data binary        |27.01 &pm; 10.09|14.23 &pm; 9.64 |
> | 1024 JSON binary          |48.86 &pm; 1.42 |32.20 &pm; 6.93 |
> | 1024 JSON MCQA heuristic  |49.50 &pm; 1.35 |42.38 &pm; 0.54 |
>
> As shown in the table, both approaches significantly improve upon the few-shot baseline. Specifically, our method with the heuristic achieves an F1 score of 49.50 ± 1.35 on ARC-Easy and 42.38 ± 0.54 on ARC-Challenge. Interestingly, this heuristic-based approach outperforms direct binary classification (48.86 ± 1.42 and 32.20 ± 6.93 F1 on ARC-Easy and ARC-Challenge, respectively). We hypothesize that this is because the heuristic, by leveraging the full probability distribution from the MCQA training, better captures the model's confidence in its predictions compared to a simple binary classification approach.
>
> Furthermore, our framework can be applied to more general QA tasks, such as scoring candidate answers retrieved from a knowledge base. In this scenario, the LLM could generate question-answer pairs, and our method could train an efficient encoder-only model to score the plausibility of retrieved answers. This approach offers advantages in terms of efficiency and scalability compared to using the LLM directly for scoring, particularly when dealing with a large number of candidate answers.

---

> ### Author Response · Authors · 2024-11-21
> **Response to Reviewer U1vS(Part 2)**
>
> > W3. The model does not outperform Tasksource model which is obtained by the multi-task training of the same backbone: the improvement on MMLU is marginal (+0.5), and on ARC data the proposed approach works significantly worse.
>
> While the average improvement of 0.5 on MMLU appears marginal, it's important to consider that this is an aggregated result across 57 diverse datasets. Furthermore, the improvement on certain subsets, such as STEM (+1.6), is more substantial. For the ARC tasks, fine-tuning the Tasksource model with our JSON-distilled data yields significant gains, which we shown on table below.
>
>
> | Method 		    | ARC-Easy | ARC-Challenge 	|
> |---------------------------|----------|----------------|
> | Tasksource 		    | 72.8     | 51.2 		|
> | Decompose distill 	    | 67.8     | 45.3 		|
> | JSON distill 		    | 69.8     | 48.6 		|
> | Tasksource + JSON distill | 74.5     | 54.7  		|
>
>
> We achieve an accuracy of 74.5% on ARC-Easy and 54.7% on ARC-Challenge. The JSON-distill approach, even without leveraging Tasksource's multi-task training, achieves performance close to the Tasksource baseline, highlighting the effectiveness of our data generation and distillation method. Our method achieves this performance using only a limited number of initial examples for data generation, whereas Tasksource benefits from extensive multi-task training. This demonstrates the efficiency of our method in leveraging limited real-world data.
>
> > Q1. Do you have any idea, if the huge improvement of the performance of DeBerta after distillation is related to improving of the model's question answering ability, or just due to learning of Multiple Choice QA format?
>
> To investigate whether the performance improvement comes solely from learning the MCQA format or also from improved question answering ability, we conducted the following experiment. We use the LLaMa-3.1-8B-Instruct generated 1024 ARC Easy examples with JSON data generation methods. We then trained a DeBERTa-v3-base model on this generated ARC Easy data with distillation. We compared its performance on MMLU with a model trained directly on MMLU generated data with distillation and the 5-shot baseline.
>
> |Method 		   | STEM | Social Science | Humanities | Other | Average |
> |--------------------------|------|----------------|------------|-------|---------|
> |Trained on MMLU generated | 32.5 | 43.2 	   | 44.3 	| 40.6  | 39.3    |
> |Trained on Arc-e 5-shot   | 22.0 | 22.8 	   | 21.9 	| 22.5  | 22.3    |
> |Trained on Arc-e generated| 32.3 | 40.5 	   | 41.4	| 40.3  | 37.9    |
>
> Training on the ARC Easy generated data significantly improves performance over the 5-shot baseline. However, the model trained on MMLU generated data still performs better, achieving an average accuracy of 39.3%. This gap suggests that our method is not merely teaching the model the MCQA format, but is also enabling it to acquire task-specific knowledge relevant to the MMLU datasets. Therefore, we conclude that the improvements observed from our method stem from both an improved understanding of the MCQA format and, more importantly, an enhanced ability to answer questions within the specific domains covered by MMLU.

---

> ### Author Response · Authors · 2024-12-01
> **Rebuttal Feedback Request for Reviewer U1vS**
>
> Dear Reviewers U1vS
>
> A gentle reminder regarding the rebuttal for paper 3797. The rebuttal period is ending soon, and we haven't yet received feedback from you. Your insights are greatly appreciated. We would be grateful if you could take a look at your earliest convenience. Thank you.

---

> ### Comment · Reviewer_U1vS · 2024-12-03
> **Response to authors' rebuttal**
>
> Dear authors,
> I highly appreciate your efforts and the additional experiments you performed.
>
> Still, I keep some of my concerns: (1) the novelty is limited; (2) the applicability is limited, the method underperforms standard training; (3) the improvement upon multi-task Tasksource model is not convincing.
> I wish your paper to be published at some less competitive venue.
>
> This time, I've decided to keep my scores.

---

### Note · Authors · 2024-12-13

**Comment:**

We thank the reviewers for their valuable feedback and suggestions.

**Withdrawal Confirmation:**

I have read and agree with the venue's withdrawal policy on behalf of myself and my co-authors.